# Research on Direct Yaw Moment Control of Electric Vehicles Based on Electrohydraulic Joint Action

Lixia Zhang, Taofeng Yan , Fuquan Pan * , Wuyi Ge and Wenjian Kong

School of Mechanical and Automobile Engineering, Qingdao University of Technology, No. 777 Jialingjiang Road, Qingdao 266520, China
* Correspondence: fuquanpan@yeah.net

**Abstract:** To solve the problem of lateral instability of the vehicle caused by insufficient lateral force of the tires due to the insufficient torque provided by the motor to the tire when the vehicle turns sharply or avoids obstacles in an emergency, a layered control method is used to design a lateral stability control system. The upper decision layer selects the yaw rate and the sideslip angle of the center of mass as the control variables and uses the joint state deviation of the yaw rate and the sideslip angle of the center of mass and the rate of change of the deviation as the input of the sliding mode variable structure controller to calculate the additional yaw moment required to maintain vehicle stability. The lower torque distribution layer realizes the distribution of torque through the electro-hydraulic coordinated control method: the torque distribution rule based on real-time load transfer calculates the torque corresponding to the control wheel and generates the torque through the hub motor and transmits it to the wheel. When the torque output from the motor cannot provide sufficient torque for the vehicle, hydraulic braking is used as a compensating control, and the difference between the required yaw torque and the motor-generated yaw torque is used as the required torque for hydraulic control to calculate the wheel cylinder pressure required to brake the wheels. Based on the joint simulation model of MATLAB/Simulink and Carsim, the sine and double shift line working condition are selected for stability simulation experiments. From the simulation results, it can be seen that the yaw rate and sideslip angle of the center of mass of the vehicle with sliding mode control and electro-hydraulic coordinated control almost coincide with the ideal value curve, which are both smaller than the output parameters of the uncontrolled vehicle. From the perspective of the motor output torque, compared with pure motor control, the effect of electro-hydraulic coordinated control is better, and the hydraulic system can compensate for the braking torque in time and enhance the lateral stability of the vehicle. The designed control strategy can make the yaw rate and the sideslip angle of the center of mass of the vehicle follow the reference value better, which can effectively avoid the vehicle sideslip and instability and improve the vehicle yaw stability and driving safety. However, due to the limitations of experimental equipment, the proposed method could not be applied to the real vehicle test. The real vehicle test can better test the control effect of the proposed method.

**Keywords:** yaw stability; sliding mode control; torque distribution; electrohydraulic coordinated control

## 1. Introduction

At present, the commonly used active safety systems of automobiles include anti-lock braking systems, electronic stability control systems, active front wheel steering systems and direct yaw moment control systems. According to the survey, the proportion of traffic accidents caused by the vehicle instability phenomenon gradually increases. Therefore, direct yaw moment control (DYC) has become one of the important topics of vehicle stability research. Direct yaw moment control judges the driver's steering intention by collecting the steering wheel angle signal and then distributing the driving force or braking force of the wheels to generate the yaw moment around the center of mass to regulate the yaw motion of the car to restrain the over/under steering of the car and improve the

stability of the car under extreme working conditions. The DYC system is mainly divided into the differential braking DYC system, differential driving DYC system and differential braking and differential driving integrated DYC system [1].

When the vehicle turns sharply or avoids obstacles in an emergency, it is easy to cause lateral instability of the vehicle due to insufficient lateral force of the tire, and the phenomenon of vehicle drifting and loss of control can easily lead to traffic accidents. For the lateral stability of the vehicle, it is mainly controlled by direct yaw moment [2], active front wheel steering [3,4], etc., to avoid the occurrence of vehicle instability. In this paper, combined with the research content, the direct yaw moment control method is used for research.

Yu et al. [5] have conducted a great deal of research on direct yaw moment control and proposed that vehicle stability control is mainly achieved through yaw moment decision and torque distribution. The torque decision mainly includes fuzzy control, PID, sliding mode control (SMC) and other methods to control the sideslip angle of the center of mass or the yaw rate. The torque distribution includes the optimal distribution algorithm, the average distribution method, the pseudo-inverse method and the load-based distribution method. In this paper, sliding mode control is adopted in the torque decision, and the integral term is introduced into the traditional sliding mode surface. In the torque distribution, the torque is first distributed based on the real-time load transfer of the wheels; second, when the motor output torque is insufficient, the demand torque is redistributed by hydraulic braking as the compensation control.

Yaw moment control based on fuzzy control proposed in the literature [6–8] is widely used in yaw moment control because of its strong robustness. However, the fuzzy control algorithm directly fuzzes the information and relies relatively on experience, which can reduce the control accuracy. Considering the problem of control accuracy, Kim et al. [9] used the PID control theory to calculate the direct yaw moment of the vehicle and used the differential braking torque coordinated control strategy to distribute the four-wheel torque, but the control strategy was only applicable to the low-speed case and the operating conditions with a small sideslip angle of the center of mass. The sideslip angle of the center of mass increases at high speed. Lin et al. [10] calculated the additional yaw moment based on the sliding mode control algorithm, overcame the shortcomings of fuzzy control and PID control and proposed an optimal distribution control strategy for driving torque with the purpose of improving handling stability and reducing motor energy consumption. However, sliding mode control has the phenomenon of jitter vibration. Nam et al. [11] adopted the adaptive sliding mode control method to control the sideslip angle of the center of mass of the vehicle and applied the parameter adaptive law to estimate the change in vehicle parameters with the road conditions, reducing the jitter vibration phenomenon and model uncertainty in sliding mode control. Thang et al. [12] established a yaw moment controller based on adaptive sliding mode control, and combined sliding surface and adaptive gain control law studies were derived from the errors in both the yaw rate and the sideslip angle of the center of mass of the actual and reference signals as a way to reduce the jittering phenomenon of sliding mode control, and the research results show that the system can enhance the stability of the vehicle. Zhang et al. [13] designed a feedforward controller based on driver command resolution to regulate the yaw rate gain, which improved the sliding mode control algorithm, designed an integral sliding mode controller for feedback control and tracked the desired motion of the vehicle. The results show that the proposed strategy reduces the maximum yaw rate of the vehicle to within 6% and 9% of the ideal yaw rate range. In the above studies, joint control of the yaw rate and the sideslip angle of the center of mass is less considered. When joint control is used, only a simple addition is performed, which cannot guarantee control accuracy. Then, the steady-state tracking error generated by external disturbances is less considered. In this paper, we propose to use the joint state deviation and deviation change rate of the yaw rate and the sideslip angle of the center of mass as the input of the sliding mode control algorithm and adopt the coordinated weighting coefficients to adjust the joint control variables to ensure accuracy and introduce

an integration term to reduce the steady-state error and suppress the jitter generated by the system through the method of high-gain feedback. Benefits of sliding mode control include: the response speed of sliding mode is relatively fast and requires fewer parameters to be adjusted; second, sliding mode control is not sensitive to disturbances, which is very suitable for vehicle stability control [14].

Some researchers have fixed the distribution of torque demand according to some specific rules. Generally, this type of control system cannot take full advantage of the independent controllability of the drive wheels according to the form state of the vehicle. Kawashima et al. [15] maintained the yaw moment required by the vehicle in a stable state and realized the distribution of yaw moment through the differential drive of the motor, and they followed the principle that the same side is in the same operating condition. Saikia et al. [16] applied the desired yaw moment calculated by the upper sliding mode controller to the wheels on both sides of the vehicle, and the wheels on both sides would generate a braking pressure difference so as to realize the torque on the vehicle control. The literature [17] adopts a layered control strategy for vehicle stability control. The upper layer control calculates the additional yaw moment required to correct the vehicle motion state based on sliding mode control, and the lower layer control adopts the form of unilateral differential braking to distribute the additional yaw moment. Some scholars have applied a more flexible control allocation method for DYC research. The literature [18] established an objective function for minimizing the energy consumption of the system and optimized the economy under different operating conditions under the given constraints to obtain the torque distribution value. The literature [19] designed the optimization function with the goal of the fastest maneuvering response and converted it into a quadratic function to solve the optimal solution of the drive torque distribution. This method has control accuracy only when the optimal solution exists. When the optimal solution does not exist, its allocation error is relatively large. Alcantar et al. [20] used a quadratic planning method to distribute the driving torque of the wheels based on the limit utilization rate of road adhesion. Zhang et al. [21] adjusted the yaw moment generated by reducing the braking force of one side of the wheel, but this could not guarantee the braking performance of the vehicle. Zhang et al. [22] established a yaw moment distribution method based on the ratio of axle load and distributed the yaw moment to four drive hub motors as the distribution layer and obtained experimental data through the joint simulation of Carsim and MATLAB/Simulink to verify the effectiveness of the established FSMC control method. From the perspective of driving torque distribution, Dizqah et al. [23] proposed a four-wheel drive torque distribution method based on the optimal control of the whole vehicle speed. This method effectively improves the driving stability of the independent drive electric vehicle and improves the economy of the whole vehicle during driving. Tian et al. [24] calculated the yaw moment and rear wheel angle required for vehicle steering based on the control strategy combining four-wheel steering and direct yaw moment and distributed the braking force according to the method of single-side braking. The distribution of the braking torque and the correction of the steering angle enable the vehicle's yaw rate and the sideslip angle of the center of mass to track the ideal model. In the torque distribution method, the above research ignores the constraints of motor torque characteristics, tire adhesion limit and other constraints, which have certain limitations. Second, most of the research is based on the traditional vehicle to achieve the control of yaw moment by means of differential braking or differential driving without considering the combination of differential braking and differential driving to optimize the control of yaw moment. This combination method is more conducive to optimal torque distribution, ensuring that each wheel can provide sufficient torque and reduce energy loss.

In electric vehicles, the hydraulic braking system can generate a large torque, but the response is slow, while the motor torque response is fast, but the output capacity is limited [25]. The literature [26] designed a layered electrohydraulic composite control strategy. First, a robust adaptive slip rate controller was designed to solve the total braking torque, and then it was distributed by an optimization-based control distribution strategy,

taking into account the position constraint and rate constraint of the motor and hydraulic system as well as the charging and discharging rate constraint of the battery. Wu et al. [27] proposed that the energy efficiency of a powertrain can be improved by the torque distribution between the front and rear wheels under normal driving conditions. Under extreme driving conditions, an electric motor combined with a hydraulic brake system was used as the actuator for direct yaw moment control. Ou et al. [28] used sliding mode control to calculate the required yaw moment to maintain vehicle stability and adopted differential braking to distribute the required yaw moment through the electrohydraulic control system. The simulation results show that the control effect of the yaw rate and the center of mass slip angle is very good and the anti-instability ability of the vehicle is improved.

In terms of electrohydraulic control, most of the studies compare the torque distribution of motor characteristics and the torque distribution method of hydraulic differential braking. The motor torque distribution is used as the main distribution strategy, while the hydraulic braking, as the compensation control distribution of the joint action torque distribution strategy, is less studied. Because the motor has the characteristics of fast response and precise control, its motor peak torque is not very large; in contrast, the hydraulic braking system can provide larger braking torque, but its braking response is slow and the control algorithm is complex. Therefore, this paper combines motor torque distribution with hydraulic brake torque distribution to give full play to the advantages of motor and hydraulic braking and improving the braking efficiency of the vehicle but also making up for the shortcomings of the traditional braking method and ensuring that the vehicle is in a balanced state when turning, improving ride comfort and safety. On the premise of meeting the stability requirements, the economy of the vehicle is also considered to solve the problem of the vehicle losing stability due to insufficient motor output torque.

In this paper, we choose electric vehicles as the research object. In the stability control strategy, we consider the advantages of motor torque control, and the yaw moment distribution control strategy based on electrohydraulic joint action is proposed. The upper decision layer designs the sliding mode variable structure controller based on the joint state deviation of the yaw rate and the sideslip angle of the center of mass, uses coordinated weighting coefficients to regulate both, selects the high-gain feedback method to suppress chattering and enables the control system to stabilize quickly by adjusting the parameter values. The lower distribution layer realizes the distribution of torque by combining the motor differential drive and differential brake based on the load distribution method. When the motor output torque is insufficient, the hydraulic brake is used as the compensation control according to the difference between the demanded yaw moment and the motor-generated yaw moment, and this control method combining the motor and hydraulic brake can ensure that the vehicle is always in a stable driving state when turning. Finally, based on the joint simulation model under sinusoidal operating conditions, the effectiveness of the stability controller and the torque distribution strategy is verified.

## 2. Vehicle Dynamics Model

### 2.1. Two Degrees of Freedom Vehicle Dynamics Model

To study the lateral motion characteristics of the vehicle, it is only necessary to consider the influence of the vehicle lateral motion and yaw motion on the vehicle driving stability [29]. Therefore, a two degrees of freedom vehicle dynamics model is established as a reference model to describe the state of the vehicle, and corresponding control is applied according to the vehicle motion state. The two degrees of freedom vehicle dynamics model is shown in Figure 1.

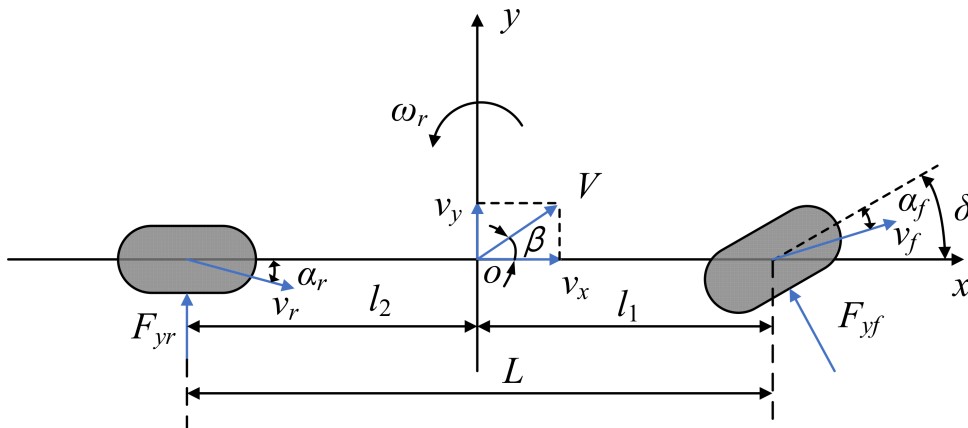

**Figure 1.** Two degrees of freedom vehicle dynamics model.

According to the force analysis, the force balance equation along the $y$ axis and around the $z$ axis can be obtained as follows:

$$\begin{cases} m(\dot{v}_y + v_x \cdot \omega_r) = F_{yf}\cos\delta + F_{yr} \\ I_z\dot{\omega}_r = l_1 \cdot F_{yf}\cos\delta - l_2 \cdot F_{yr} \end{cases} \tag{1}$$

It is assumed that all angles are small and that tire cornering characteristics are considered to be linear. The differential equation of motion for a two degrees of freedom vehicle can be obtained as follows:

$$\begin{cases} m(\dot{v}_y + v_x \cdot \omega_r) = \left(k_f + k_r\right)\beta + \frac{1}{v_x}\left(l_1 k_f - l_2 k_r\right)\omega_r - k_f\delta \\ I_z\dot{\omega}_r = \left(l_1 k_f - l_2 k_r\right)\beta + \frac{1}{v_x}\left(l_1^2 k_f + l_2^2 k_r\right)\omega_r - l_1 k_f\delta \end{cases} \tag{2}$$

where $m$ is the vehicle mass, $v_x$ is the longitudinal vehicle speed, $v_y$ is the lateral vehicle speed, $\omega_r$ is the yaw rate, $\beta$ is the sideslip angle of the center of mass, $F_{yf}$ is the lateral force of the front wheel, $F_{yr}$ is the lateral force of the rear wheel, $\delta$ is the wheel steering angle, $I_z$ is the moment of inertia around the $z$ axis, $l_1$ is the distance from the center of mass to the front axle, $l_2$ is the distance from the center of mass to the rear axle, $k_f$ is the cornering stiffness of the front wheel and $k_r$ is the cornering stiffness of the rear wheel.

### 2.2. Vehicle Model Based on Carsim

The vehicle model in Carsim is selected as the actual control model, but the software does not develop a motor model for electric vehicles, so MATLAB/Simulink is used to build the motor model and cut off the power transmission between the drive train and the wheels in Carsim. The motor was connected to the vehicle model in Carsim by setting the parameter interface to perform motor power control in the vehicle model.

The B-Class Hatchback vehicle model in Carsim software is selected as the whole vehicle model of the control system, and the transmission system is modified in the software. The rest of the vehicle body parameters, tire parameters, aerodynamic parameters, etc., continued to use the parameters set by the software, and the main parameters of the vehicle model are shown in Table 1.

As the actuator of the control system, the motor controls the output of power, and the stability of the vehicle is determined by the performance of the motor. According to the relationship between the performance of the motor and the vehicle dynamics index, the selected motor must meet the vehicle dynamics performance requirements. The vehicle dynamics indices studied in this paper are shown in Table 2.

**Table 1.** Main parameters of the vehicle model.

| Parameter | Value | Symbol |
|---|---|---|
| Vehicle mass (kg) | 1235.0 | $m$ |
| Wheel base (mm) | 2600.0 | $L$ |
| Distance from the center of mass to the front axle (mm) | 1040.0 | $l_1$ |
| Distance from the center of mass to the rear axle (mm) | 1560.0 | $l_2$ |
| Front/rear wheel tread (mm) | 1480.0 | $B_f/B_r$ |
| Height of center of mass (mm) | 540.0 | $h$ |
| Wheel radius (mm) | 357.0 | $R$ |
| Front wheel cornering stiffness (N·rad$^{-1}$) | −79,240.0 | $k_f$ |
| Rear wheel cornering stiffness (N·rad$^{-1}$) | −87,002.0 | $k_r$ |
| Moment of inertia around z axis (kg·m$^2$) | 1343.1 | $I_z$ |

**Table 2.** Dynamic performance index of the vehicle.

| Vehicle Dynamics Index | Value |
|---|---|
| Maximum speed (km/h) | 160 |
| Maximum grade (%) | 30 (20 km/h) |
| 100 km acceleration time (s) | <10 |

By matching and calculating various parameters of the motor, the basic parameters of the motor are obtained as shown in Table 3.

**Table 3.** Basic parameters of the motor.

| Motor Parameters | Value |
|---|---|
| Rated power (kW) | 10 |
| Peak power (kW) | 25 |
| Rated torque (N·m) | 120 |
| Peak torque (N·m) | 370 |
| Rated speed (r/min) | 800 |
| Peak speed (r/min) | 1500 |

The permanent magnet brushless DC motor has the characteristics of high stability, small size and high efficiency [30], and it is selected as a hub motor. The external characteristic curve of the motor is shown in Figure 2. Due to the low torque and the high power characteristics of the motor, the output torque of the motor remains unchanged at low speed and the output torque decreases with increasing speed at high speed, but the output power remains unchanged. Then, the relationship between the motor torque, speed and power is expressed as follows:

$$T = \begin{cases} T_e & ,0 < n \le n_e \\ \frac{9549 P_e}{n} & ,n_e < n \le n_{\max} \end{cases} \tag{3}$$

where $T$ is the actual output torque of the motor, $T_e$ is the rated torque of the motor, $P_e$ is the rated power, $n$ is the motor speed, $n_e$ is the rated speed of the motor and $n_{\max}$ is the peak speed.

In this paper, we study the stability control method of the vehicle: the accuracy requirement of the motor is not high, and the output torque by the table look-up method is used to meet the requirements of the external characteristics of the motor. The motor model built based on MATLAB/Simulink is shown in Figure 3.

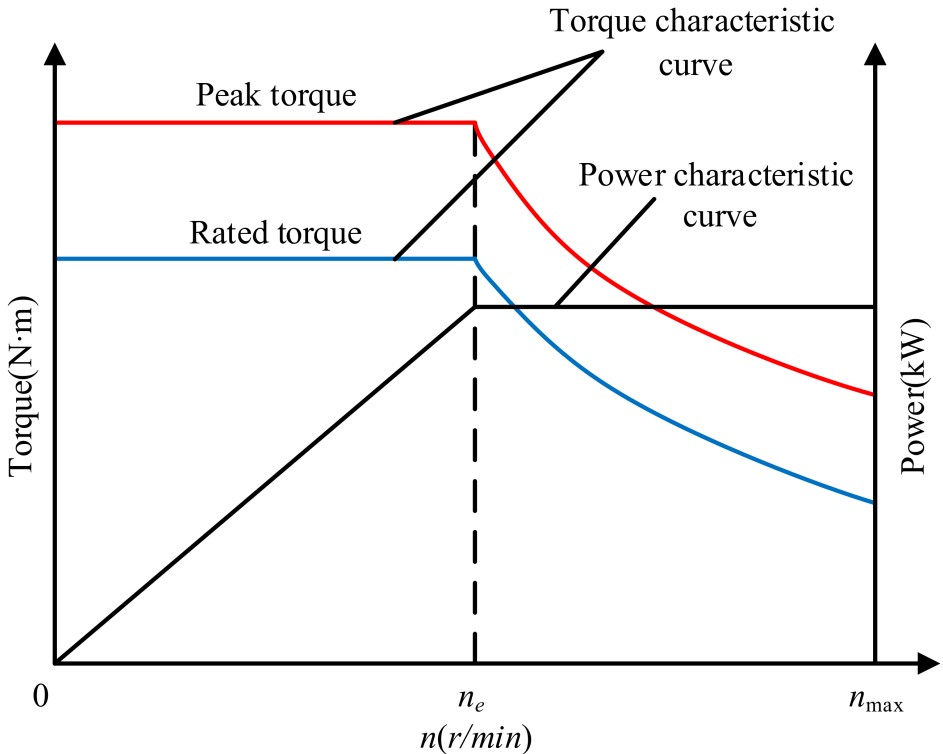

**Figure 2.** The external characteristic curve of the motor.

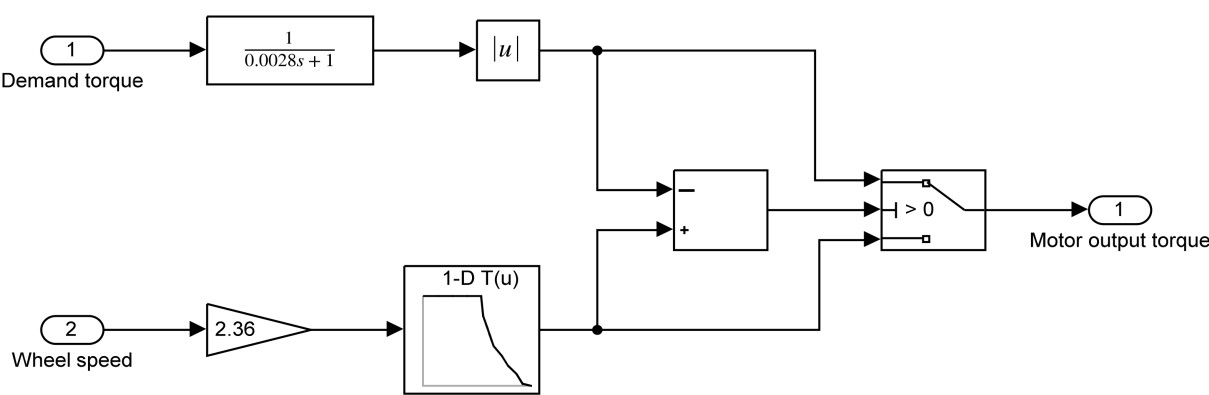

**Figure 3.** Simulink motor models.

## 3. Research on Yaw Moment Control Decision

### 3.1. Design of the Yaw Moment Control Strategy Scheme

In order to solve the problem that the motor cannot provide enough torque for the tires when the vehicle turns sharply or avoids obstacles in an emergency, resulting in insufficient lateral force of the tires, causing the lateral instability of the vehicle and then a the occurrence of a serious traffic accident, this paper proposes a hierarchical control structure scheme for the lateral stability control strategy of the vehicle, as shown in Figure 4. The overall scheme design includes a vehicle stability judgment module, yaw moment control module, lower layer moment distribution module and actuator. The vehicle stability judgment module and the direct yaw moment controller based on sliding mode control form the upper decision layer; the lower torque distribution module and the actuator form the lower torque distribution layer. The interaction between the various modules improves the lateral stability of the vehicle and enhances the driving stability of the vehicle.

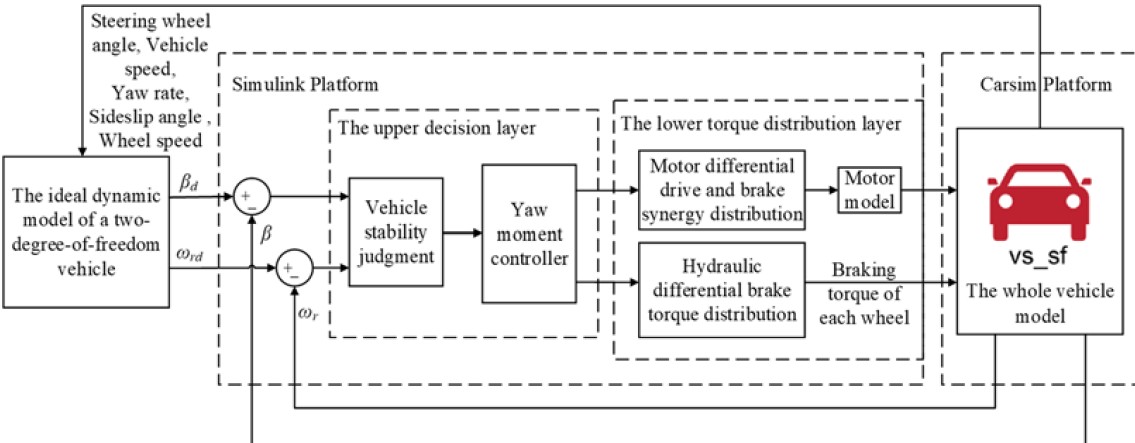

**Figure 4.** Yaw moment control structure diagram.

### 3.2. Vehicle Stability Judgment

The yaw rate and the sideslip angle of the center of mass are important indicators to measure the stability of the vehicle. The yaw rate is mainly used to determine whether the car will understeer or oversteer during the steering process, and the sideslip angle of the center of mass can be used to determine whether there will be trajectory deviation during the steering process. When the sideslip angle of the center of mass is small, the yaw rate can characterize the stability of the vehicle. However, when the sideslip angle of the center of mass is large, the yaw rate makes it difficult to measure the stability of the vehicle. At this time, it is particularly important to control the sideslip angle of the vehicle [31]. Based on this, this paper selects the yaw rate and the center of mass sideslip to cooperate with each other to jointly determine the stability state of the vehicle.

When judging the instability of the vehicle, the coupling effect of the yaw rate and the sideslip angle of the center of mass should be considered, and the influence condition of the yaw rate and the sideslip angle of the center of mass on the vehicle stability should be analyzed in detail, so the driving state of the vehicle is evaluated by combining the phase plane method of the sideslip angle of the center of mass—the sideslip angle velocity of the center of mass $\beta - \dot{\beta}$ and the $\omega_r$ threshold method, and the judgment process is shown in Figure 5.

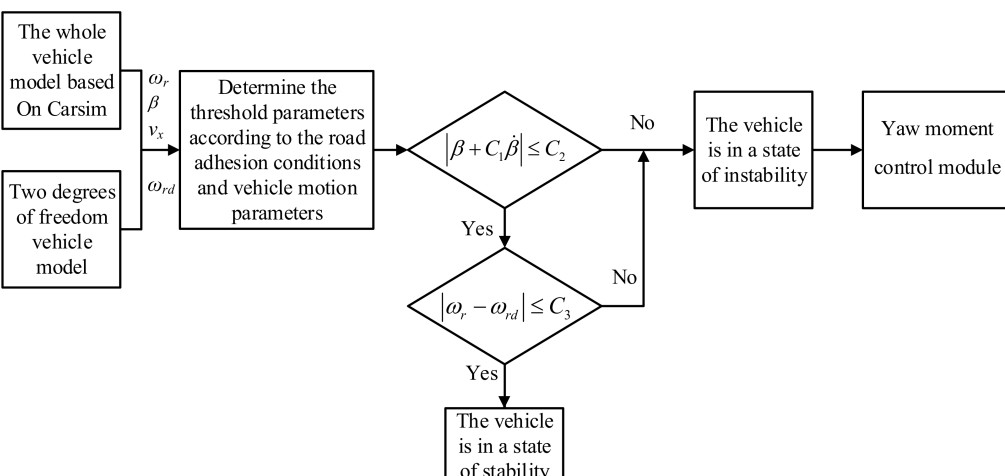

**Figure 5.** Lateral stability controller control logic.

Stability judgment criteria are: according to the vehicle dynamics model, to obtain the motion state parameters, such as the yaw rate and the sideslip angle of the center of mass. According to Table 4, determine the values of $C_1$ and $C_2$, and substitute them into the

formula $\left| \beta + C_1\dot{\beta} \right| \leq C_2$ for calculation. If the formula is not satisfied, it indicates that the vehicle is unstable and requires yaw moment control; otherwise, it is necessary to calculate whether the yaw rate deviation exceeds the critical value $C_3$. If the yaw rate deviation exceeds the critical value, it indicates that the vehicle is unstable and needs to be controlled. If the yaw rate deviation is within the critical value, it indicates that the vehicle is stable and does not require control.

**Table 4.** Stability domain boundary parameters.

| Number | The Road Surface Adhesion Coefficient $\mu$ | $C_1$ | $C_2$ |
| --- | --- | --- | --- |
| one | $\mu < 0.2$ | 0.284 | 2.577 |
| two | $0.2 \leq \mu < 0.4$ | 0.297 | 3.345 |
| three | $0.4 \leq \mu < 0.6$ | 0.303 | 4.228 |
| four | $0.6 \leq \mu < 0.8$ | 0.357 | 4.654 |
| five | $0.8 \leq \mu < 1.0$ | 0.357 | 5.573 |

From Figure 5, it can be seen that the boundary constant of the stability domain is determined according to the road adhesion coefficient, and then two motion state parameters of the yaw rate and the sideslip angle of the center of mass are obtained from the whole vehicle model to determine whether the sideslip angle of the center of mass of the vehicle satisfies the condition. The yaw rate is judged for the vehicle that satisfies the condition, the vehicle that does not satisfy the two conditions is in the unstable domain and stability control needs to be applied. $C_3$ is the limit value of the yaw rate, which is related to factors such as the road surface adhesion coefficient and speed, and its value is determined by reference to the literature [32]. The values of the stability domain boundary parameters $C_1$ and $C_2$ are shown in Table 4.

### 3.3. Ideal Vehicle Dynamics Model

When the vehicle is in a state of stability, the yaw rate $\omega_r$ remains constant, that is, $\dot{v}_y, \dot{\omega}_r = 0$, which is substituted into Formula (2), and, considering the road adhesion conditions and the actual condition of the tire under the ultimate working condition, the ideal yaw rate and the ideal sideslip angle of the center of mass can be deduced by the two degrees of freedom vehicle dynamics model as follows:

$$\begin{cases} \omega_{rd} = \min\left\{ \left| \frac{v_x\delta}{L(1+Kv_x^2)} \right|, \left| \xi \cdot \frac{\mu g}{v_x} \right| \right\} \cdot \text{sgn}(\omega_r) \\ \beta_d = \min\left\{ \left| \frac{v_x^2\delta}{L(1+Kv_x^2)}\left( \frac{l_2}{v_x^2} + \frac{ml_1}{k_rL} \right) \right|, \left| \tan^{-1}(0.02\mu g) \right| \right\} \cdot \text{sgn}(\beta) \end{cases} \tag{4}$$

where $K = \frac{m}{L^2}\left( \frac{l_1}{c_r} - \frac{l_2}{c_f} \right)$ is the stability factor, $L = l_1 + l_2$ is the wheel base, $\text{sgn}(x)$ is the sign function, $\xi$ is the stability coefficient at the ultimate working condition, taken as 0.85, and $\mu$ is the road surface adhesion coefficient.

### 3.4. Design of the Yaw Moment Controller

Sliding mode control is essentially nonlinear control, which can change purposefully and continuously according to the current state of the system in the dynamic process, forcing the system to move in accordance with the predetermined "sliding mode" state trajectory. In addition, the sliding mode control is not sensitive to perturbations. When designing the controller, there are few parameters to be adjusted, and the response speed is fast, which has the advantage of good control of the stability of the vehicle [33]. This paper is based on the two degrees of freedom vehicle dynamics model. The upper decision layer calculates the ideal yaw rate and the ideal slip angle of the center of mass according to the vehicle's speed and steering angle. Taking the deviation of the yaw rate and the deviation of the sideslip angle of the center of mass as the input quantities of the joint control, the required additional yaw moment is calculated by the sliding mode control algorithm. The

lower torque distribution layer distributes the required additional yaw moment according to the result calculated by the decision layer. The torque corresponding to the control wheels is calculated according to the torque distribution rules based on real-time load transfer, the wheel motor is made to generate the corresponding amount of torque to be transferred to the wheels so that the torque of the left and right wheels are different and the hydraulic brake is used as a compensating control to redistribute the demanded torque when the output torque of the motor is insufficient. Through the combination of motor torque control and hydraulic brake compensation control, the direct yaw moment control of the vehicle is enabled to ensure the driving stability of the vehicle. The control decision structure is shown in Figure 6.

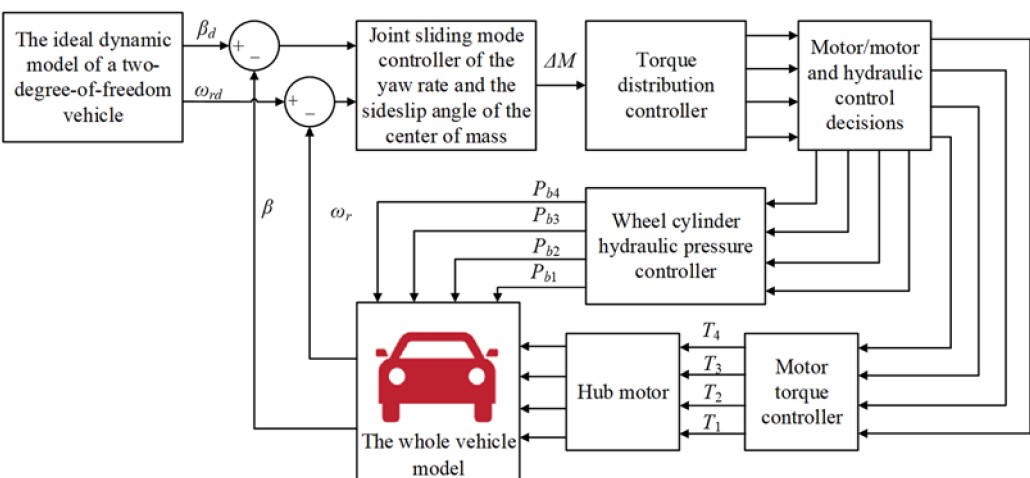

**Figure 6.** The decision diagram of the yaw moment on sliding mode control.

According to the ideal model of the vehicle, the output is the ideal yaw rate $\omega_{rd}$ and the sideslip angle of the center of mass $\beta_d$, and the sliding mode variable structure controller is designed with the deviation $e$ and the deviation change rate $\dot{e}$ of the yaw rate and the sideslip angle of the center of mass. Considering the coupling relationship between the yaw rate and the sideslip angle of the center of mass, the coordinated weighting coefficients are designed to control the proportion of the two, which can achieve better stability control.

The two degrees of freedom vehicle dynamics model is an ideal car model, so the equation for applying the additional yaw moment $\Delta M$ is as follows:

$$\begin{cases} m(\dot{v}_y + v_x\omega_r) = F_{yf} + F_{yr} \\ I_z\dot{\omega}_r = l_1 F_{yf} - l_2 F_{yr} + \Delta M \end{cases} \tag{5}$$

Define the joint state deviation of the variables as follows:

$$e = (\omega_r - \omega_{rd}) + \lambda(\beta - \beta_d) \tag{6}$$

where $\lambda$ is the coordinated weighting coefficient for the sideslip angle of the center of mass.

By analyzing the causes of vehicle instability, the expression of the weighting coefficient $\lambda$ is determined as follows:

$$\lambda = \begin{cases} 1 & |\beta| \geq \beta_1 \\ 1 - \frac{|\beta| - \beta_0}{\beta_1 - \beta_0} & \beta_0 < |\beta| < \beta_1 \\ 0 & |\beta| \leq \beta_0 \end{cases} \tag{7}$$

where $\beta_0$ is the lower limit of the sideslip angle of the center of mass and $\beta_1$ is the upper limit of the sideslip angle of the center of mass [34].

For the whole system, the external disturbance will lead to steady-state tracking error. To reduce the steady-state error and improve the tracking accuracy, an integral term $\int_0^t edt$ is introduced into the traditional sliding surface to obtain the integral sliding surface as follows:

$$s = \dot{e} + \lambda_1 e + \lambda_2 \int_0^t edt \tag{8}$$

where $\lambda_1$ and $\lambda_2$ are sliding surface coefficients, and $\lambda_1$ and $\lambda_2 > 0$ are constants.

Taking the derivative of Formula (8) and combining Formula (6), we can obtain:

$$\dot{s} = \left[ (\ddot{\omega}_r - \ddot{\omega}_{rd}) + \lambda \left( \ddot{\beta} - \ddot{\beta}_d \right) \right] + \lambda_1 \left[ (\dot{\omega}_r - \dot{\omega}_{rd}) + \lambda \left( \dot{\beta} - \dot{\beta}_d \right) \right] \\ + \lambda_2 [(\omega_r - \omega_{rd}) + \lambda(\beta - \beta_d)] \tag{9}$$

To ensure that the sliding mode system performs the sliding mode, the designed sliding mode variable structure controller adopts the control method of the exponential approach law:

$$\dot{s} = -\varepsilon \text{sgn}(s) - ks \quad k, \varepsilon > 0 \tag{10}$$

where $k$ is the gain coefficient, $\varepsilon$ is the controller parameter and chattering occurs when the actual state of the system approaches the sliding mode surface. To suppress the chattering of the system, $k$ can be appropriately increased and the value of $\varepsilon$ can be decreased in the design, so choosing a reasonable $\varepsilon$ and $k$ can ensure the quality of the system approaching the sliding mode surface, which is conducive to rapid stabilization of the system.

Deformation of Formula (2), the differential equation for applying an additional yaw moment, is obtained:

$$\begin{cases} \dot{\omega}_r = \frac{l_1 k_f - l_2 k_r}{I_Z} \beta + \frac{l_1^2 k_f + l_2^2 k_r}{I_Z v_x} \omega_r - \frac{l_1 k_f}{I_Z} \delta + \frac{\Delta M}{I_Z} \\ \dot{\beta} = \frac{k_f + k_r}{m v_x} \beta + \left( \frac{l_1 k_f - l_2 k_r}{m v_x^2} - 1 \right) \omega_r - \frac{k_f}{m v_x} \delta \end{cases} \tag{11}$$

Taking the derivative of Formula (11) and substituting it into Formula (9), we obtain:

$$\begin{aligned} \dot{s} &= \left( \frac{l_1 k_f - l_2 k_r}{I_z} + \lambda \frac{k_f + k_r}{m v_x} + \lambda \lambda_1 \right) \dot{\beta} + \left[ \frac{l_1^2 k_f + l_2^2 k_r}{I_z v_x} + \lambda \left( \frac{l_1 k_f - l_2 k_r}{m v_x^2} - 1 \right) + \lambda_1 \right] \dot{\omega}_r \\ &\quad - \frac{l_1 k_f}{I_z} \dot{\delta} - \lambda \frac{k_f}{m v_x} \delta - \ddot{\omega}_{rd} - \lambda \ddot{\beta}_d - \lambda_1 \dot{\omega}_{rd} - \lambda \lambda_1 \dot{\beta}_d + \lambda_2 (\omega_r - \omega_{rd}) \\ &\quad + \lambda \lambda_2 (\beta - \beta_d) + \frac{\Delta \dot{M}}{I_z} \\ &= Q + \frac{\Delta \dot{M}}{I_z} \end{aligned} \tag{12}$$

Combining Formula (10) and Formula (12) and integrating them, the additional yaw moment required for the vehicle to maintain stability can be obtained as:

$$\Delta M = -I_z \int (\varepsilon \text{sgn}(s) + ks + Q) dt \tag{13}$$

The stability of the designed sliding mode variable structure control is proved, and the Lyapunov function is used to judge the stability of the system. Define the Lyapunov function as:

$$V = \frac{1}{2} s^2 \tag{14}$$

The derivative of Formula (13) can be:

$$\dot{V} = s\dot{s} \tag{15}$$

Since $k, \varepsilon > 0$, according to Formulas (10), (12) and (15), the system is stable.

Since the ideal switching characteristics do not exist in the sliding mode control law, which leads to easy generation of chattering during the switching of the system, and to weaken the chattering generated by the system, the method of high-gain feedback is chosen

to suppress the chattering. The method chooses the nonlinear function $v_\delta(s)$ to replace the symbolic function $\text{sgn}(s)$ in the controller, and its expression is as follows:

$$v_\sigma(s) = \frac{s}{|s| + \sigma}, \sigma > 0 \tag{16}$$

where $|s|$ is the norm of $s$, $\sigma$ is the parameter and adjusting the parameter $\sigma$ can make the state variables reach the sliding mode surface quickly.

## 4. Research on the Distribution Strategy of Yaw Moments

When the vehicle loses its stability, four motors are used to act simultaneously to apply different degrees of braking torque and driving torque to the four wheels to generate the required additional yaw moment, thereby adjusting the driving state of the vehicle. First, the vehicle is controlled by a combination of motor differential drive and differential braking. When the motor torque is insufficient, hydraulic braking is used as compensation control. Therefore, a yaw moment distribution strategy based on the combined action of the electric motor and hydraulic pressure is proposed.

### 4.1. Distribution Strategy of Motor Torque

The additional yaw moment $\Delta M$ calculated by the sliding mode controller is distributed to the four wheels in a coordinated way of driving and braking. To improve the driving stability of the vehicle, according to the distribution strategy of increasing torque to one wheel and reducing torque to the other wheel, and in accordance with the driver's driving intention and the motion state of the vehicle, select the wheel that needs to apply torque [35]. In this paper, the front wheel angle is set to be positive when turning left and positive when counterclockwise. The specific distribution strategy is shown in Table 5.

**Table 5.** Driving and braking cooperation distribution strategy.

| $e_\omega = \omega_r - \omega_{rd}$ | Front Wheel Angle δ | Direct Yaw Moment | State of the Vehicle | Brake Wheel | Drive Wheels |
|---|---|---|---|---|---|
| $e_\omega > 0$ | $\delta > 0$ | negative | oversteer | Reduced torque on the right side of the wheel | Increased torque on the left side of the wheel |
| $e_\omega < 0$ | $\delta > 0$ | positive | understeer | Reduced torque on the left side of the wheel | Increased torque on the right side of the wheel |
| $e_\omega > 0$ | $\delta < 0$ | negative | understeer | Reduced torque on the right side of the wheel | Increased torque on the left side of the wheel |
| $e_\omega < 0$ | $\delta < 0$ | positive | oversteer | Reduced torque on the left side of the wheel | Increased torque on the right side of the wheel |

When performing the torque distribution, it is necessary to meet the requirement of an additional yaw moment; that is, the total longitudinal moment of the four wheels is equal to the total longitudinal demand moment of the vehicle, and the resultant moment generated by the four wheels around the center of mass of the vehicle should be equal to the total expected yaw moment. Assuming that the torques assigned to each wheel are $T_1$, $T_2$, $T_3$ and $T_4$, we obtain:

$$T_{xd} = T_1 + T_2 + T_3 + T_4 \tag{17}$$

$$\Delta M = \frac{B_f}{2R}(T_2 - T_1) + \frac{B_r}{2R}(T_4 - T_3) \tag{18}$$

According to theoretical mechanics, the torque is distributed based on the real-time load transfer of the wheels so that the vehicle has sufficient tire adhesion to overcome the

instability of the vehicle. According to Formula (19), the torque of the four wheels of the vehicle can be obtained as:

$$\begin{cases} T_1 = \dfrac{F_{z1}}{\sum\limits_{i=1}^{4} F_{zi}} \left( \dfrac{T_{xd}}{2} - \dfrac{\Delta M}{B_f} R \right) \\[3mm] T_2 = \dfrac{F_{z2}}{\sum\limits_{i=1}^{4} F_{zi}} \left( \dfrac{T_{xd}}{2} + \dfrac{\Delta M}{B_f} R \right) \\[3mm] T_3 = \dfrac{F_{z3}}{\sum\limits_{i=1}^{4} F_{zi}} \left( \dfrac{T_{xd}}{2} - \dfrac{\Delta M}{B_r} R \right) \\[3mm] T_4 = \dfrac{F_{z4}}{\sum\limits_{i=1}^{4} F_{zi}} \left( \dfrac{T_{xd}}{2} + \dfrac{\Delta M}{B_r} R \right) \end{cases} \qquad (19)$$

In addition, the torque $T_i$ of each wheel is limited by the peak torque of the motor and the road adhesion conditions [36] as follows:

$$T_i \leq \min(\mu F_{zi} R, T_{\max}) \qquad (20)$$

where $i = 1, 2, 3, 4$ are the left front wheel, right front wheel, left rear wheel and right rear wheel, respectively; $T_{xd}$ is the total longitudinal demand torque of the vehicle; $T_i$ is the torque generated by the motor of each wheel; $F_{zi}$ is the vertical load of each wheel and $T_{\max}$ is the maximum output torque of the motor.

From the above formulas, the distribution strategy of motor torque based on MAT-LAB/Simulink is shown in Figure 7.

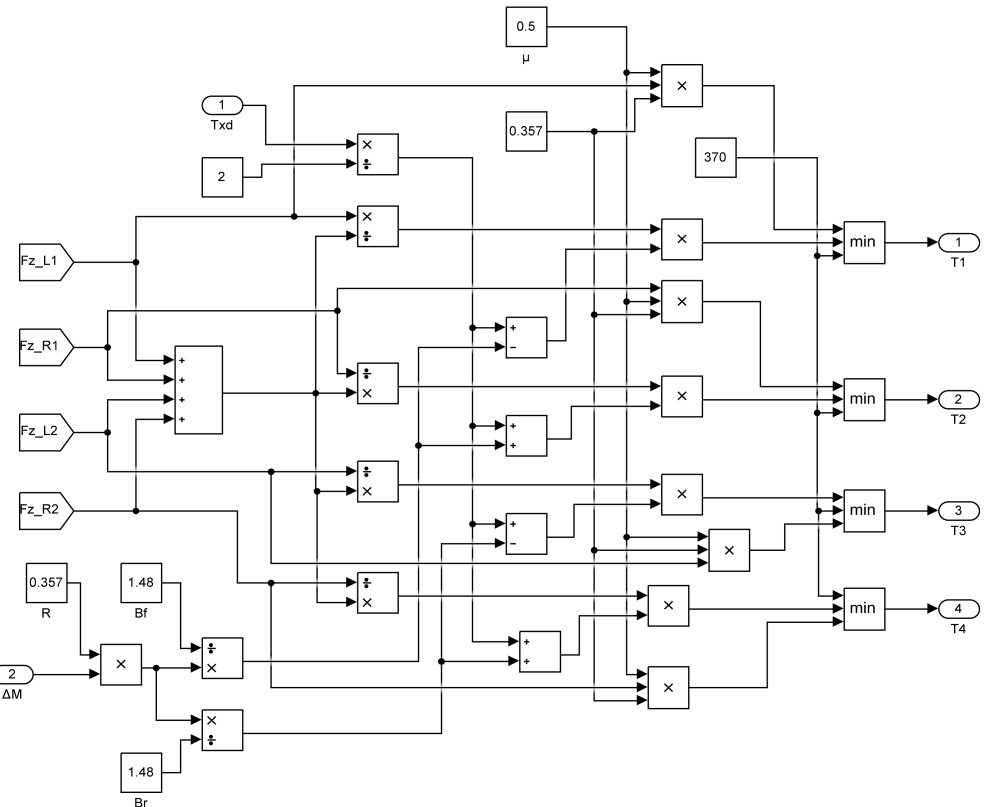

**Figure 7.** Motor torque distribution strategy diagram based on Simulink.

### 4.2. Hydraulic Brake Compensation Control Strategy

When the torque output from the motor cannot provide enough torque for the vehicle, it is necessary to start the hydraulic system to start working to compensate for the braking force control. The judgment formula is as follows:

$$T_i' = \begin{cases} T_{\max} + T_{bi} & T_i > T_{\max} \\ T_i & T_i \leq T_{\max} \end{cases} \tag{21}$$

In the case of compensation in the form of hydraulic braking, the compensation is calculated by the difference between the required yaw moment and the yaw moment generated by the motor:

$$\begin{cases} \Delta M_Z = (T_2 - T_1) \cdot \frac{B_f}{2R} + (T_4 - T_3) \cdot \frac{B_r}{2R} \\ \Delta M_H = \Delta M - \Delta M_Z \end{cases} \tag{22}$$

where $\Delta M_H$ is the yaw moment generated by hydraulic brake compensation control and $\Delta M_Z$ is the maximum yaw moment generated by the motor output torque.

According to the yaw moment $\Delta M_H$ required for hydraulic braking, the single-wheel braking method is selected when the hydraulic braking force is distributed. The wheel cylinder pressure corresponding to the braking wheel and the compensating torque generated by hydraulic braking are calculated as:

$$P_{bi} = \frac{2\Delta M_H R}{B \cdot r \cdot A \cdot K_b} \tag{23}$$

$$T_{bi} = K_b \cdot P_{b_i} \tag{24}$$

where $B$ is the wheel tread, $r$ is the effective radius of the brake, $A$ is the effective area of the brake, $T_{bi}$ is the compensating braking torque generated by the hydraulic brake, $K_b$ is the braking efficiency factor from the brake wheel cylinder pressure to the wheel braking torque and $P_{bi}$ is the wheel cylinder pressure of the brake wheel.

In the case of hydraulic braking as compensation control, the single-wheel braking method is selected to achieve torque distribution, and the control rules are based on the input change of the front wheel angle and the change range of the difference between the actual value and the ideal value of the yaw rate to determine which wheel the car should control. The hydraulic differential braking torque distribution rules are shown in Table 6.

**Table 6.** Hydraulic differential braking torque distribution strategy.

| $e_\omega = \omega_r - \omega_{rd}$ | Front Wheel Angle δ | Direct Yaw Moment | State of the Vehicle | Brake Wheel |
|---|---|---|---|---|
| $e_\omega > 0$ | δ > 0 | negative | oversteer | Right front wheel |
| $e_\omega < 0$ | δ > 0 | positive | understeer | Left rear wheel |
| $e_\omega > 0$ | δ < 0 | negative | understeer | Right rear wheel |
| $e_\omega < 0$ | δ < 0 | positive | oversteer | Left front wheel |

The flow chart of motor and hydraulic brake compensation control is shown in Figure 8. The lower torque distribution layer distributes the additional yaw moment. First, the torque is distributed through the motor torque controller to obtain the torque of the four wheels, and then the control is applied to the wheels to be controlled according to the distribution strategy. When the output torque of the motor cannot provide enough torque for the wheels, hydraulic brake compensation is used. The hydraulic brake compensation control strategy is shown in Figure 9.

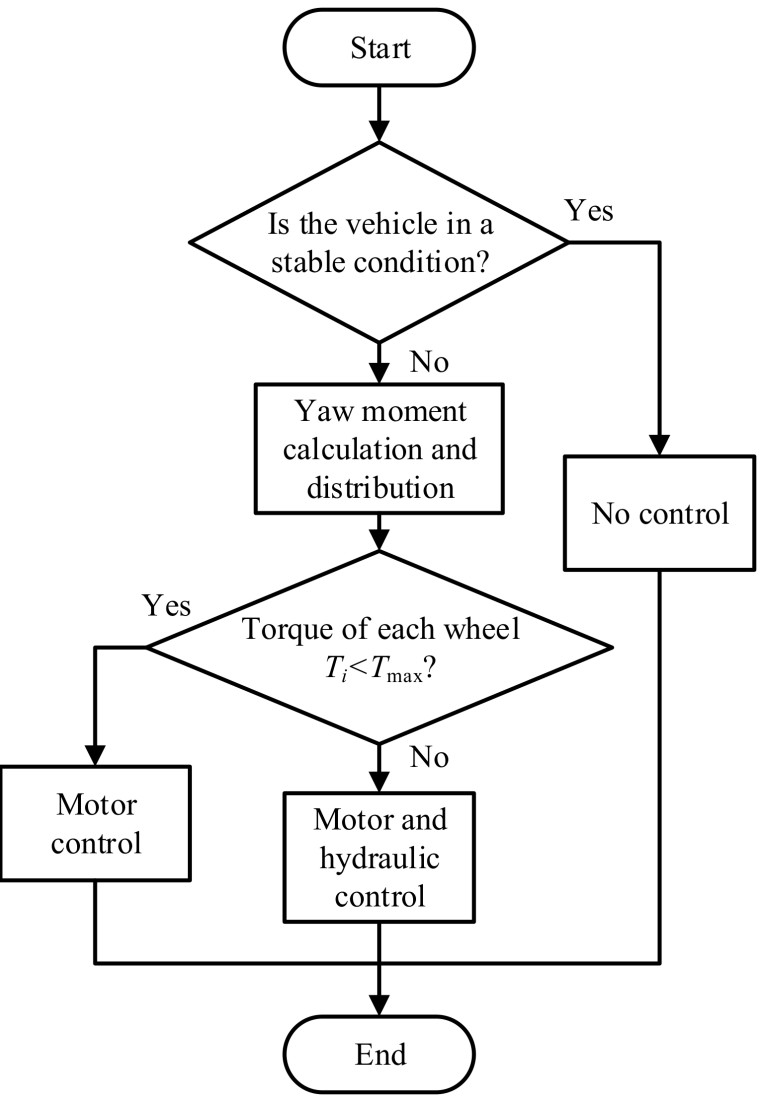

**Figure 8.** Motor and hydraulic brake compensation control flow chart.

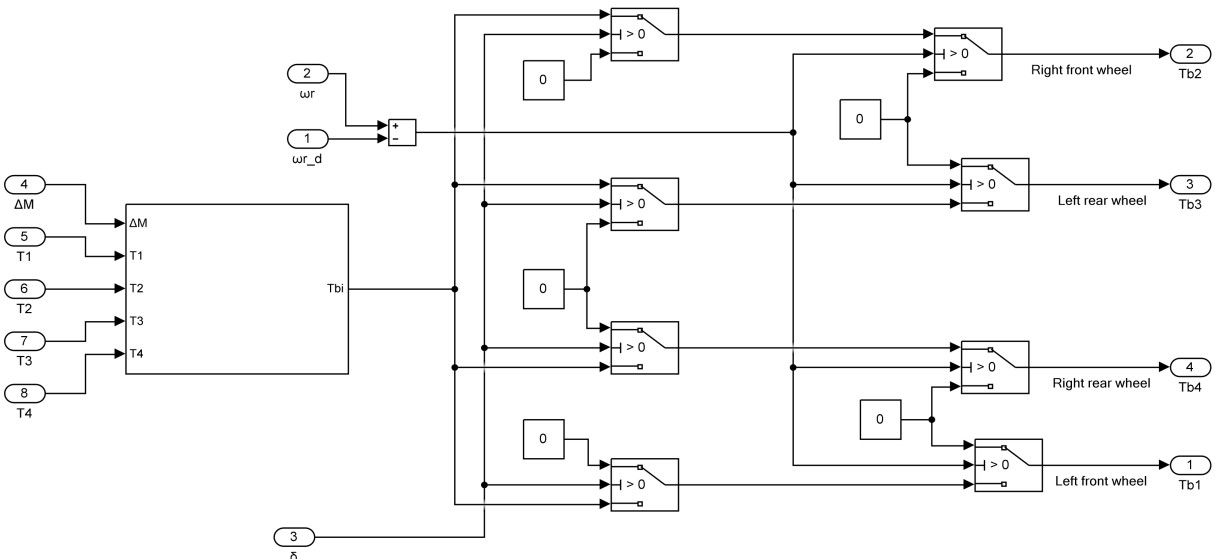

**Figure 9.** Hydraulic brake compensation control strategy diagram.

Through the above formulas, the torque distribution strategy diagram of motor and hydraulic coordinated control is established, as shown in Figure 10.

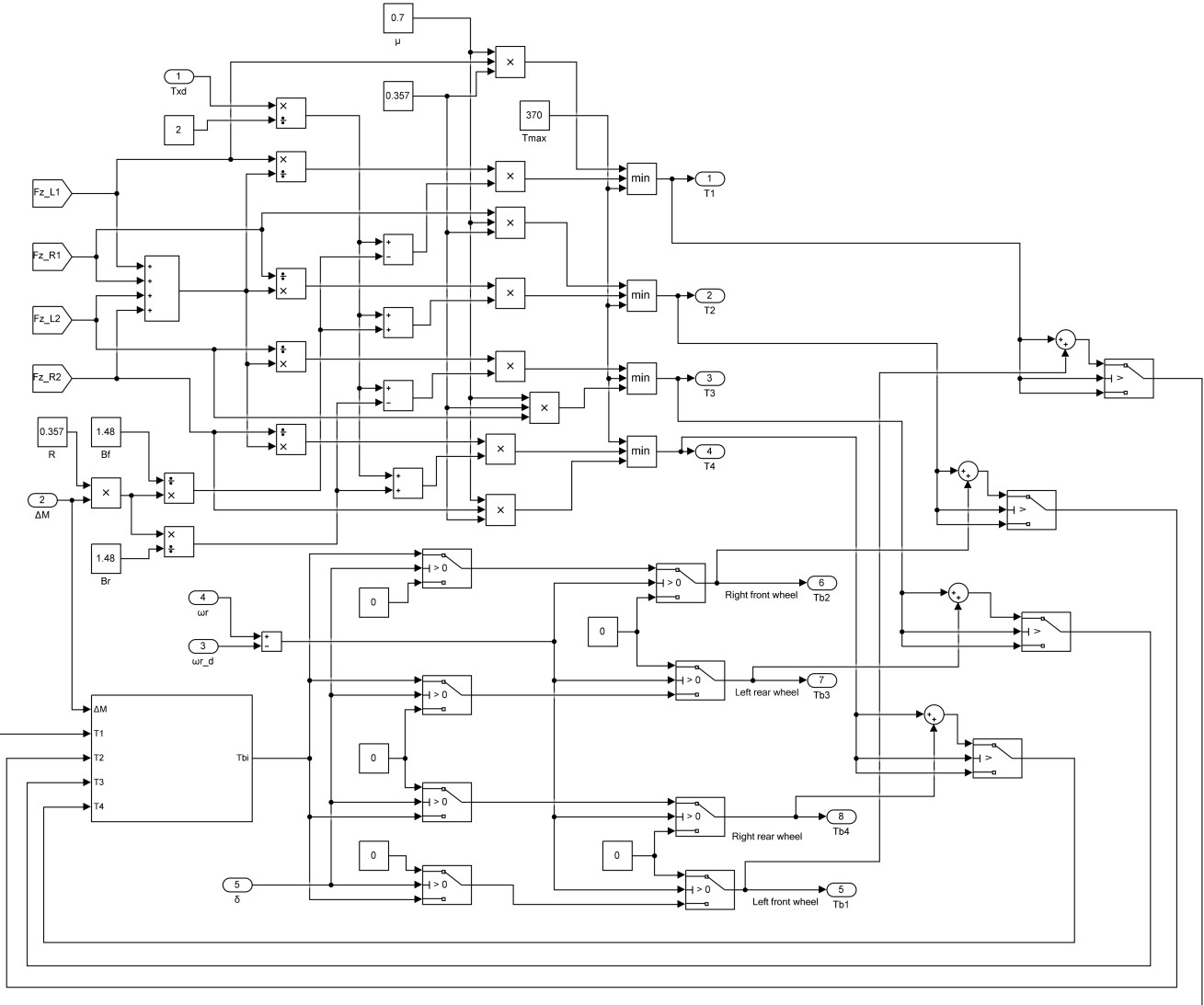

**Figure 10.** Torque distribution strategy diagram of motor and hydraulic coordinated control.

## 5. Stability Simulation Analysis of Yaw Moment Control

In this paper, the established model and controller are jointly simulated based on MATLAB/Simulink and Carsim software. Based on the whole vehicle simulation model, the entire control system model is built, and the whole vehicle joint simulation model based on electrohydraulic coordinated control is established, as shown in Figure 11.

### 5.1. Sine Condition Test

To verify the control effect of the designed system, the sine and the double-lane change line working condition are selected for simulation. In the sinusoidal condition, the settings are: the initial vehicle speed is 80 km/h, the road adhesion coefficient is 0.7 and the front wheel angle input signal is shown in Figure 12. Based on the joint simulation model for simulation verification, the values of vehicle parameters under ideal, no control and applied control effects are compared, and the simulation results are shown in Figures 13–18.

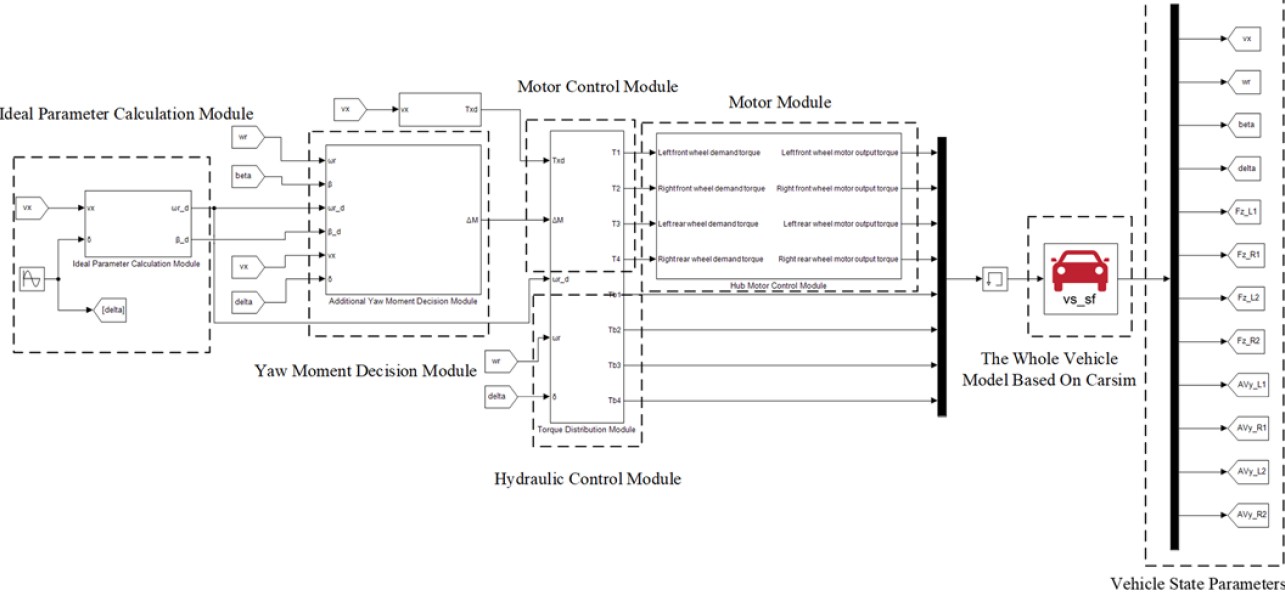

**Figure 11.** Carsim–Simulink co-simulation model.

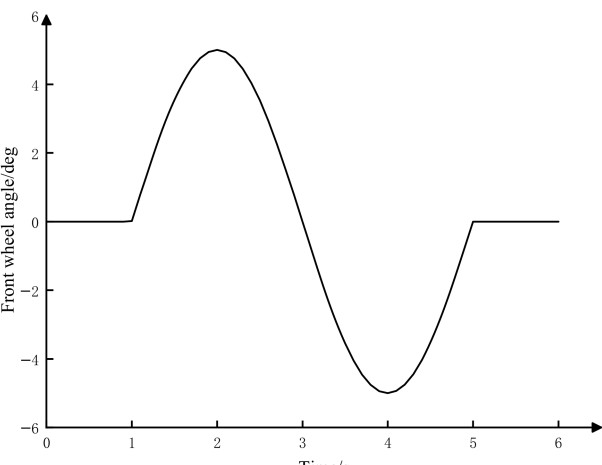

**Figure 12.** Front wheel angle of vehicle in the sinusoidal condition.

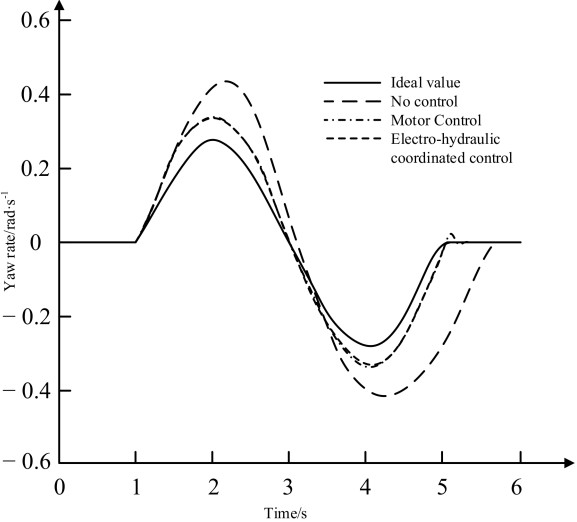

**Figure 13.** The curve of the yaw rate in the sinusoidal condition.

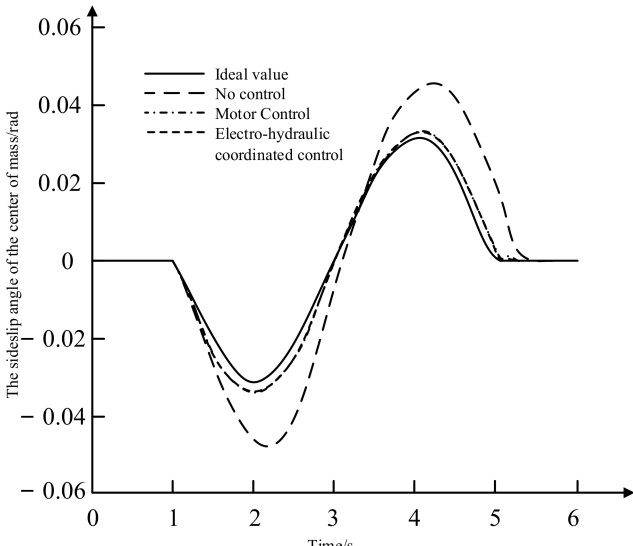

**Figure 14.** The curve of the sideslip angle in the sinusoidal condition.

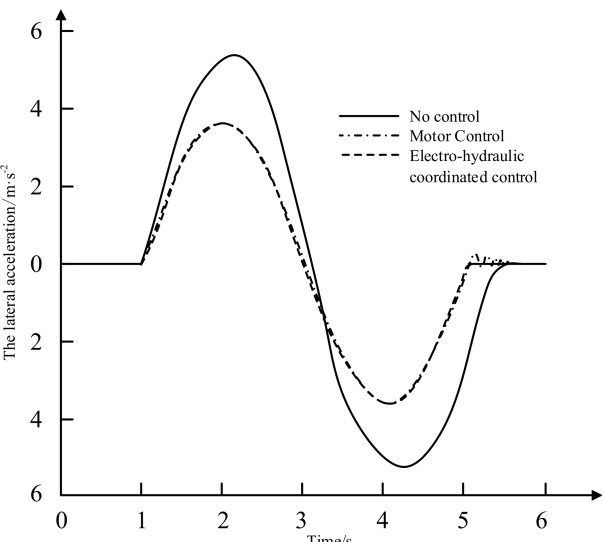

**Figure 15.** The curve of lateral acceleration in the sinusoidal condition.

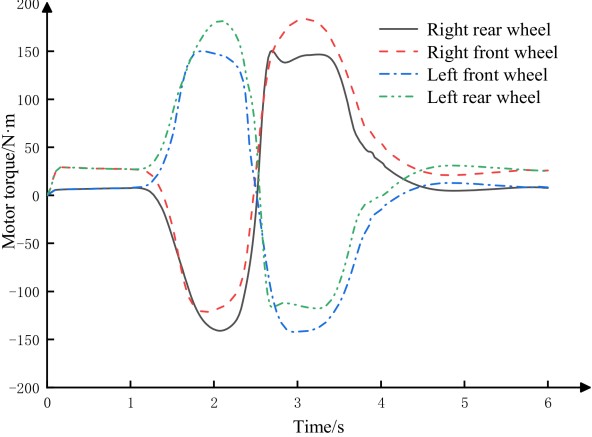

**Figure 16.** Motor torque curve in the sinusoidal condition (motor control).

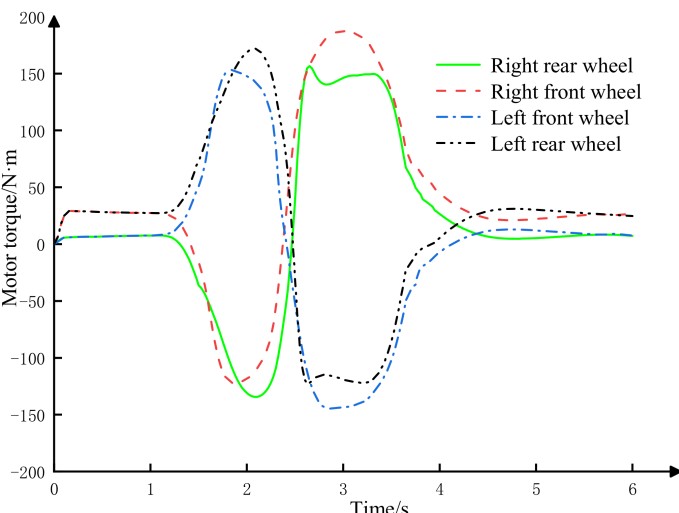

**Figure 17.** Motor torque curve in the sinusoidal condition (electrohydraulic coordinated control).

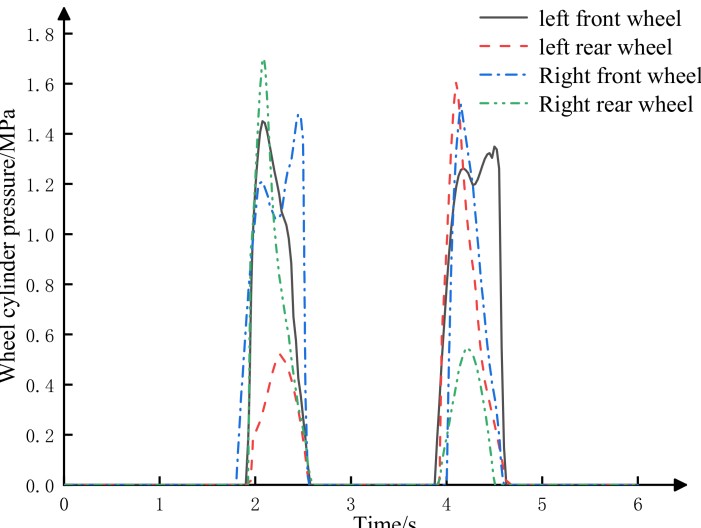

**Figure 18.** Wheel cylinder pressure curve in the sinusoidal condition (electrohydraulic coordinated control).

Figures 13 and 14 show that the yaw rate and sideslip angle of the center of mass of the vehicle without control have a large change range; their values deviate greatly from the ideal values, and the stability of the vehicle is poor at this time. However, the vehicle with pure motor control and electrohydraulic coordination control is closer to the ideal value of 0.28 rad/s and 0.032 rad. Among them, the peak values of pure motor control are about 0.32 rad/s and 0.035 rad. Compared with pure motor control, the electrohydraulic coordinated control reduced by about 0.018 rad/s and 0.002 rad, and, at the same time, compared with no control, the peak yaw rate of the vehicle with control is reduced by 0.10 rad/s, an increase of about 24%, and the peak of the sideslip angle of the center of mass is reduced by 0.012 rad and improved by 27%. Obviously, the yaw rate and the sideslip angle of the center of mass curve of the controlled vehicle have a small change range, which is consistent with the ideal value curve, and can track the ideal value curve well. However, the yaw rate curve and the sideslip angle of the center of mass curve of the electro-hydraulic coordinated control are closer to the ideal curve, and the control effect is good.

Figure 15 shows that the lateral acceleration is as high as 5.2 m/s$^2$ when the vehicle is not controlled, and the vehicle enters the steady state for a longer time, while the lateral acceleration of the vehicle with motor control and electrohydraulic coordinated control

is 3.2 m/s$^2$, with the acceleration peak value differing by approximately 38%, and the time to enter the steady state is shorter. The vehicle with motor control shows larger fluctuations when entering the steady state, but the vehicle using the electrohydraulic coordinated control has almost no fluctuation. Compared with pure motor control, the effect of electrohydraulic coordinated control is better.

Figures 16–18 show that, before 1.2 s, the vehicle was running in a straight line, and the motor output a stable torque. At this time, the hydraulic system did not provide a braking force. At 1.2 s, the vehicle starts to turn, the motor output torque gradually increases but is not yet saturated and the hydraulic brake compensation control system is not activated at this time. As the steering wheel angle continues to increase, in approximately 2 s, the yaw rate of the vehicle reaches its peak and a sufficiently large torque must be applied to generate the required direct yaw moment. The output torque of the motor is saturated, the hydraulic compensation control system intervenes and the hydraulic pressure of the brake wheel cylinder increases. At approximately 2.2 s, the hydraulic pressure output of the brake wheel cylinder reaches a peak value of 1.68 Mpa. Starting at 2 s, the vehicle turns to the right, the motor continues to output torque and the hydraulic pressure begins to gradually decrease. At approximately 4 s, the yaw rate of the vehicle reaches its peak value, the output torque of the motor is saturated, the hydraulic compensation control system starts and the hydraulic pressure of the brake wheel cylinder rises. At approximately 4.2 s, the hydraulic pressure output of the brake wheel cylinder reaches a peak value of 1.6 Mpa. At approximately 5 s, the steering ends, the vehicle enters a stable state and continues to drive, the motor outputs a stable torque and the wheel cylinder pressure is 0 MPa. Judging from the output torque of the motor, the compensation braking torque effect of the electrohydraulic coordinated control is better.

### 5.2. Double-Lane Change Working Condition Test

In the double-lane change working condition, the settings are: the initial vehicle speed is 80 km/h, the road adhesion coefficient is 0.6, other parameters are the same as the sinusoidal condition and the front wheel angle input signal is shown in Figure 19. The simulation results are shown in Figures 20–25.

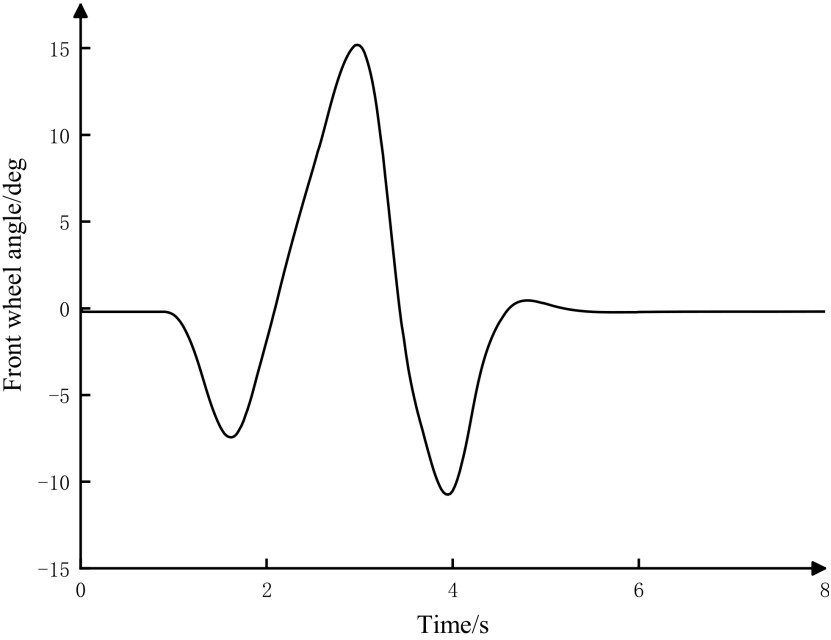

**Figure 19.** Front wheel angle of vehicle in the double-lane change working condition.

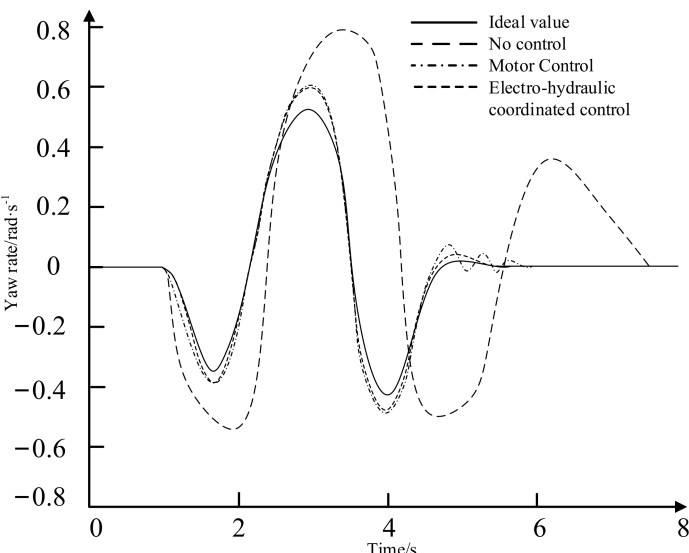

**Figure 20.** The curve of the yaw rate in the double-lane change working condition.

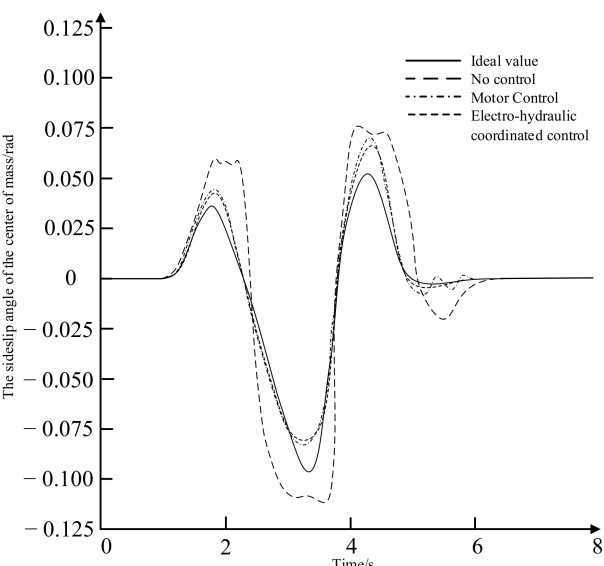

**Figure 21.** The curve of the sideslip angle in the double-lane change working condition.

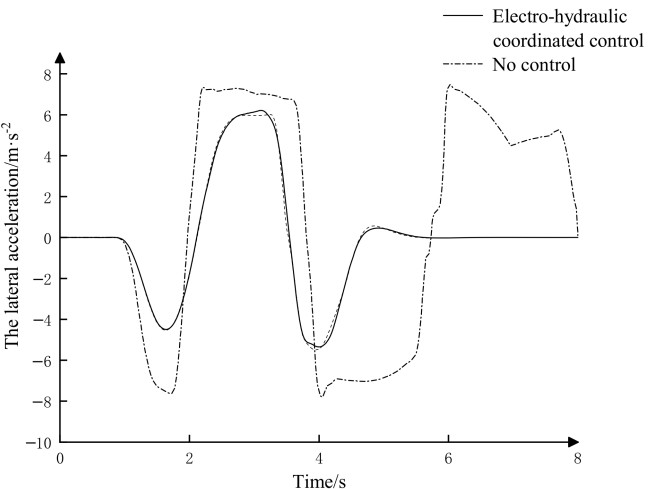

**Figure 22.** The curve of lateral acceleration in the double-lane change working condition.

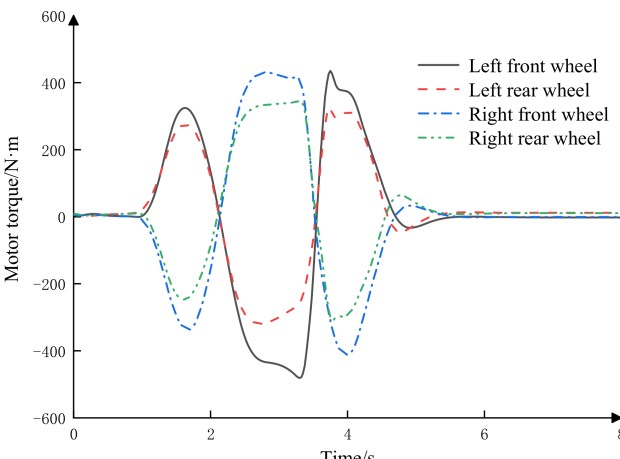

**Figure 23.** Motor torque curve in the double-lane change working condition (motor control).

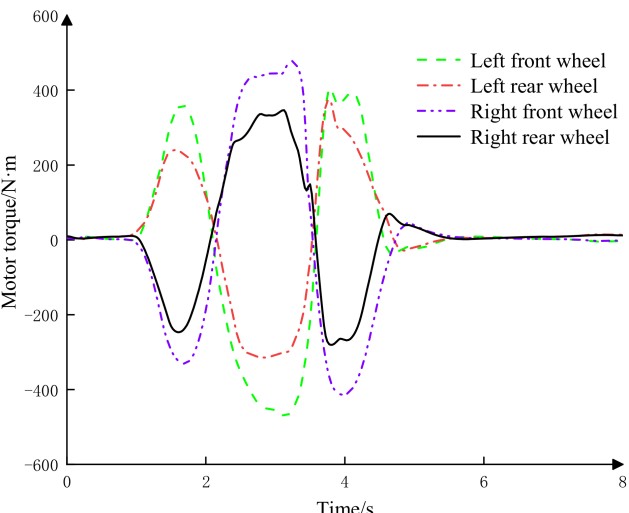

**Figure 24.** Motor torque curve in the double-lane change working condition (electrohydraulic coordinated control).

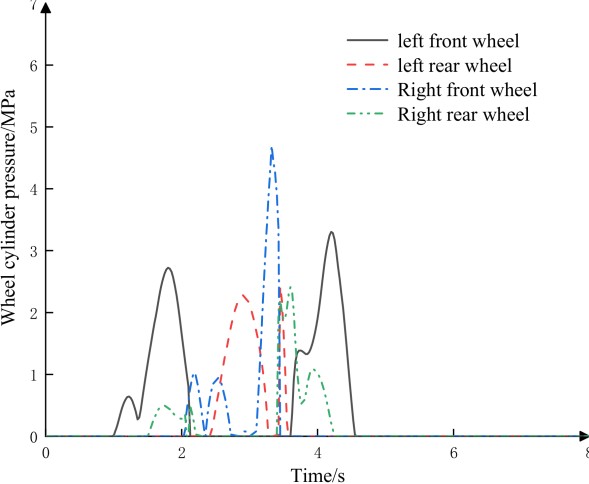

**Figure 25.** Wheel cylinder pressure curve in the double-lane change working condition (electrohydraulic coordinated control).

As can be seen from Figures 20–22, the uncontrolled vehicle experienced severe sideslip. When steering was performed, an unstable and dangerous situation occurred. At this time, the yaw rate, the sideslip angle of the center of mass and the lateral acceleration

of the vehicle increased rapidly. The amplitude of the yaw rate reached 0.8 rad/s, the maximum value of the sideslip angle of the center of mass was close to 0.075 rad and the lateral acceleration reached 7.5 m/s$^2$. However, the controlled vehicle can run stably.

It can be seen from the yaw rate curve that the amplitude of the controlled vehicle is 0.59 rad/s, which differs from the ideal value by 0.12 rad/s; it can be seen from the sideslip angle of the center of mass curve that the controlled center of the sideslip angle of the center of mass is in the range of −0.075 rad–0.03 rad, and the maximum value differs from the ideal value by 0.01 rad. In the reverse direction, the effect of applying control is better than that of the expected value. After calculation, the control effect is increased by 21%; from the lateral acceleration curve, it is known that the maximum lateral acceleration of the vehicle with the applied control is 5.8 m/s$^2$, and the effect of the applied control is better compared with the unapplied control. However, the vehicle with pure motor control has a large overshoot when entering a stable state, while the vehicle with electro-hydraulic coordinated control has almost no overshoot. In comparison, the motor control is not so effective in suppressing the overshoot of the sideslip angle of the center of mass, and the use of sliding mode control and electro-hydraulic coordinated control can reduce the lag time between the vehicle yaw rate and the reference value so that the actual vehicle yaw rate is closer to the ideal value.

As can be seen from Figures 23–25, at about 1.8 s, 3 s and 4 s, the hydraulic compensation control system intervenes, the hydraulic pressure of the brake wheel cylinder increases and the hydraulic pressure can reach up to 4.6 MPa. The effect of the electro-hydraulic coordinated control for compensating braking torque is better, and the braking torque is increased by about 18%, which is a great improvement for the performance of the vehicle and can significantly enhance vehicle stability under sharp turns. In contrast, the designed electro-hydraulic coordinated control torque distribution method is better than the pure motor control torque distribution method.

## 6. Discussion

This paper proposes a yaw moment control and distribution strategy for electric vehicles. First, the combination of the phase plane method and the threshold value method is used to evaluate the driving state of the vehicle; second, the stability control strategy of the vehicle is studied, and the additional yaw moment is calculated by using the sliding mode control algorithm; finally, the torque distribution is realized by the electro-hydraulic coordinated control system.

In addition, the wheel hub motor is used as the driving motor, and the combination of motor differential drive and differential braking and hydraulic braking is used as the compensation control method. The advantage is that each wheel can be independently driven and braked and can give full play to the respective advantages of motor braking and hydraulic braking to achieve joint braking, which is a current research hotspot, and this also makes up for the shortcomings of traditional braking methods.

On the basis of the above research, the following research can be carried out in the future:

(1) The relevant vehicle state parameters, such as the yaw rate and sideslip angle of the center of mass, are, in this paper, obtained through the Carsim vehicle model, and the vehicle state parameters are not estimated. In the future, the Kalman filtering algorithm can be added to estimate the vehicle state parameters.

(2) Due to the limitations of experimental equipment, the proposed method could not be applied to the real vehicle test, and the control effect of the proposed method can be better tested by the real vehicle test.

## 7. Conclusions

The research on the yaw stability control strategy of the vehicle can improve the driving safety of the vehicle, ensure the vehicle has good controllability and reduce traffic accidents caused by the loss of stability. The additional yaw moment distribution method

with an electrohydraulic coordination control strategy can make the vehicle better track the ideal yaw rate and the sideslip angle of the center of mass curves, improve the driving performance of the vehicle, increase the lateral stability of the vehicle and improve the stability of the vehicle under extreme working conditions.

1.  Through vehicle model establishment, motor parameterization matching, yaw moment control, torque distribution control and joint simulation, the yaw rate and the sideslip angle of the center of mass of the vehicle controlled by electro-hydraulic coordination are smaller than the output parameters of the vehicle without control applied, the yaw rate of the vehicle can better track the ideal yaw rate and the sideslip angle of center of mass can be kept in a small range and improved by about 27%, improving the vehicle's ability to follow the desired path.

2.  Compared with pure motor control, a vehicle using electrohydraulic coordinated control does not show large fluctuations when entering a steady state, the time to enter a stable state is reduced and it can quickly enter a stable state, ensuring sufficient stability when the vehicle is turned. It can correct the body orientation in time, correct the vehicle trajectory and avoid vehicle sideslip destabilization and improve the vehicle handling stability.

3.  In extreme working conditions, pure motor control is limited by the limitation of the maximum output torque of the motor, and the hydraulic brake compensation control system intervenes in time to perform auxiliary braking so that the vehicle can turn in time and continue to maintain driving stability. Compared with pure motor control, the compensation braking torque effect of electrohydraulic coordination control is good. The torque distribution strategy of electrohydraulic coordinated control can provide sufficient demand torque to solve the problem of insufficient control torque when the vehicle is turning and maintain the vehicle in a stable driving state.

**Author Contributions:** Conceptualization, L.Z. and T.Y.; methodology, L.Z.; software, T.Y.; validation, L.Z. and T.Y.; formal analysis, T.Y. and F.P.; investigation, T.Y. and W.K.; data curation, T.Y.; writing—original draft preparation, L.Z. and T.Y.; writing—review and editing, L.Z., T.Y. and W.G.; supervision, F.P.; project administration, L.Z. and F.P.; funding acquisition, L.Z. All authors have read and agreed to the published version of the manuscript.

**Funding:** This research was funded by the National Natural Science Foundation of China (No. 51505244) and Key R & D project of Shandong Province (No. 2018GGX105009) and China Postdoctoral Science Foundation Funded Project (No. 2016M590626).

**Institutional Review Board Statement:** Not applicable.

**Informed Consent Statement:** Not applicable.

**Data Availability Statement:** The data presented in this study are available on request from the author.

**Conflicts of Interest:** The authors declare no conflict of interest.

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
