# Peer review of "Research on Direct Yaw Moment Control of Electric Vehicles Based on Electrohydraulic Joint Action"

_sustainability, doi:10.3390/su141711072_

Round 1
Reviewer 1 Report
The article describes a lateral stability control system that integrates motor-generated yaw torque and hydraulic braking. This method develops an upper layer based on an SMC for the DYC system. In normal conditions, the DC-motor based torque is used in the lower layer. At rest, the hydraulic braking method is coupled with DC-motor torque to provide the desired torque provided in the upper layer. Some MATLAB/CarSim simulations have been conducted to show the performance of the proposed method. As such, the article seems to have some sounds. However, significant improvements need to be considered.
1) (The most critical challenge): The paper lacks more references and citations on this topic. For example, the article's novelty concerning other papers, such as SMC-based DYC, should be discussed. What is the novelty of using the SMC for the control system design of the DYC?
2) Fig.7 is not a standard presentation. It would be better to show the schematic of your distribution strategy.
3) (The second critical challenge): The central system you have designed in the lower layer is a distribution method that you have used to provide the desired torque that is calculated in the upper layer. Now I have a question why have you chosen to transfer the desired torque to the braking torque or DC-motors torque? I think the desired torque should be transferred to the wheels force, then provided another controller to provide the braking torque and etc.
Author Response
Response to Reviewer Comments
Dear Reviewer,
The manuscript “Research on direct yaw moment control of electric vehicles based on electrohydraulic joint action” has been revised. Firstly, we would like to express our gratitude to you for us to improve the quality of this manuscript. According to the comments, we have revised the paper, in which the modifications are highlighted in red color.
Best regards,
Taofeng Yan
Point 1: The paper lacks more references and citations on this topic. For example, the article's novelty concerning other papers, such as SMC-based DYC, should be discussed. What is the novelty of using the SMC for the control system design of the DYC?
Response 1: We appreciate your comment. In the introduction section, we have revised the Introduction section. Many references are cited on topics such as: sliding mode control, torque distribution, electro-hydraulic coordinated control. The novelty and shortcoming of the cited literature are discussed, and the research contents and methods of this paper are proposed. Pointing out the novelty of using sliding mode control for yaw moment control, and the benefit of using electro-hydraulic coordinated control.
Point 2: Fig.7 is not a standard presentation. It would be better to show the schematic of your distribution strategy.
Response 2: We appreciate your comment. Your suggestion is very good. In this paper, a schematic diagram of the overall distribution strategy for the electro-hydraulic coordinated control has been supplemented. We have added the corresponding parts in section 4 of the manuscript
Point 3: The central system you have designed in the lower layer is a distribution method that you have used to provide the desired torque that is calculated in the upper layer. Now I have a question why have you chosen to transfer the desired torque to the braking torque or DC-motors torque? I think the desired torque should be transferred to the wheels force, then provided another controller to provide the braking torque and etc.
Response 3: We appreciate your comment. Your question is very good. What I want to express is that the additional yaw moment calculated by the yaw moment control is distributed according to the torque distribution strategy, and the calculated wheel torque is transmitted to the wheel by the hub motor. But there is no motor model for electric vehicles developed in the Carsim software, so we need to establish an independent motor model outside the Carsim software, and we cannot directly input the calculated wheel torque into Carsim. If we input it directly into the Carsim, it becomes the torque provided by the engine. And in Figure 10 we also indicate for the motor model.

Reviewer 2 Report
I am remarking upon the manuscript bears the title "Research on direct yaw moment control of electric vehicles based on electrohydraulic joint action":
1- In the abstract, the authors claim the designed sliding mode controller and electrohydraulic coordinated control torque distribution strategy greatly improve the driving safety and controllability of the vehicle. How did the authors demonstrate this statement in their work? How much does greatly mean? It is true that their control system is more effective than the case without a control system but if it is significant in comparison with other control systems like the pure motor control defined in the paper?
2- The Introduction section will need to be revised. There is a paragraph as long as two pages, pages 2 and 3, which is very uncommon and vague! The literature review should not just report the conclusions of others, it is expected to know how the literature review is closely aligned with the goals of the present study.
3- In line 40, the abbreviation of DYM for direct yaw moment should be stated in line 39 where it was first used. It is the same for sliding mode control (SMC) in line 68, while first emerging in line 63.
4- In figure 5, the authors are asked to explain how they defined the yaw stability criteria. Why is the stability criterion for the sideslip angle separated from the stability criterion for the yaw rate? Shouldn't be these two criteria coupled?
5- Why has the road surface adhesion coefficient been divided into 5 ranges in table 4?
6- What are the labels of axes in figure 16?
7- In line 485, the difference in the peak value of the sideslip angle compared with an uncontrolled vehicle is 24%, authors should clarify how much this number is for the yaw rate. How much is the difference between the two controllers? Pure motor controller and electrohydraulic coordinated controller?
8- In line 529, "... 27% higher than that of not applying control ...", it is higher or lower?
9- Why did the authors select the road adhesion coefficient equal to 0.7 for stability simulation analysis? It is suggested to have the control system tested for some other road adhesion coefficients too as this parameter plays a decisive role in the yaw stability criteria.
10- Some typos:
line 78: propose ====> proposed
line 150: is ===> was
line 320: control controller ====> controller
Author Response
Response to Reviewer Comments
Dear Reviewer,
The manuscript “Research on direct yaw moment control of electric vehicles based on electrohydraulic joint action” has been revised. Firstly, we would like to express our gratitude to you for us to improve the quality of this manuscript. According to the comments, we have revised the paper, in which the modifications are highlighted in red color.
Best regards,
Taofeng Yan
Point 1: In the abstract, the authors claim the designed sliding mode controller and electrohydraulic coordinated control torque distribution strategy greatly improve the driving safety and controllability of the vehicle. How did the authors demonstrate this statement in their work? How much does greatly mean? It is true that their control system is more effective than the case without a control system but if it is significant in comparison with other control systems like the pure motor control defined in the paper?.
Response 1: We appreciate your comment. This paper is verified by simulation in Section 5,compared with no control, the peak yaw rate of the vehicle with control is reduced by 0.10rad/s, an increase of about 24%, and the peak of the sideslip angle of the center of mass is reduced by 0.012 rad and improved by 27%. Obviously, the yaw rate and the sideslip angle of the center of mass curve of the controlled vehicle have a small change range, which is consistent with the ideal value curve, and can track the ideal value curve well. But compared with the pure motor control, the electro-hydraulic coordinated control reduced by about 0.018rad/s and 0.002rad, the yaw rate curve and the sideslip angle of the center of mass curve of the elec-tro-hydraulic coordinated control are closer to the ideal curve, and the control effect is good. Secondly, the vehicle with pure motor control has a large overshoot when entering a stable state, while the vehicle with electro-hydraulic coordinated control has almost no overshoot. In comparison, the motor control is not so effective in suppressing the overshoot of the side-slip angle of the center of mass, and the use of sliding mode control and electro-hydraulic coordinated control can reduce the lag time between the vehicle yaw rate and the reference value, so that the actual vehicle yaw rate is closer to the ideal value. Findly, When the output torque of the motor is limited, the hydraulic brake compensation control system intervenes in time to perform auxiliary braking, so that the vehicle can turn in time and continue to drive stably. The electro-hydraulic coordinated control can provide sufficient demand torque to solve the problems of insufficient control torque and vehicle instability when the vehicle is turning.
Point 2: The Introduction section will need to be revised. There is a paragraph as long as two pages, pages 2 and 3, which is very uncommon and vague! The literature review should not just report the conclusions of others, it is expected to know how the literature review is closely aligned with the goals of the present study.
Response 2: We appreciate your comment. We have revised the Introduction section,layered the original literature review into a summary, and aligned with our own research goals. The literature review is divided into the research of yaw moment decision, the research of torque distribution and the research of electro-hydraulic control.
Point 3: In line 40, the abbreviation of DYM for direct yaw moment should be stated in line 39 where it was first used. It is the same for sliding mode control (SMC) in line 68, while first emerging in line 63.
Response 3: We appreciate your comment. We have stated the abbreviation of DYC for direct yaw moment in line 44 where it first appears. At the same time, the abbreviation of SMC for sliding mode control has also been stated in line 62 where it first emerging.
Point 4: In figure 5, the authors are asked to explain how they defined the yaw stability criteria. Why is the stability criterion for the sideslip angle separated from the stability criterion for the yaw rate? Shouldn't be these two criteria coupled?
Response 4: We appreciate your comment. Stability judgment criteria: according to the vehicle dynamics model to obtain the motion state parameters such as the yaw rate and the sideslip angle of the center of mass. According to Table 4, determine the values of C1 and C2, and substitute them into the formula for calculation. If the formula is not satisfied, it indicates that the vehicle is unstable and require yaw moment control. If the formula is satisfied, calculate whether the yaw rate deviation exceeds the critical value C3, where the relationship between the C3 value and the parameters such as the driving speed and the road adhesion coefficient is determined by referring to the literature [28]. If the yaw rate deviation exceeds the critical value, it indicates that the vehicle is unstable and needs to be controlled. If the yaw rate deviation is within the critical value, it indicates that the vehicle is stable and does not require control.
The method of combining the yaw rate and the sideslip angle of the center of mass is the phase plane method of the yaw rate - the sideslip angle of the center of mass, that is the joint control method. The judgment formula is . This method can fully characterize the vehicle full working condition stability characteristics, but does not consider the influence of driving speed and the road adhesion coefficient stability reign. While this paper combines the phase plane method of the sideslip angle of the center of mass - the sideslip angle velocity of the center of mass and the threshold method to evaluate the motion state of the vehicle, and fully considering the influence of the road adhesion coefficient and vehicle speed on the stability reign. For details, please refer to the literature: Guo Jianhua. Research on electronic stability coordinated control system for double-axle vehicles. Jilin University, 2008.
Point 5: Why has the road surface adhesion coefficient been divided into 5 ranges in table 4?
Response 5: We appreciate your comment. According to the use of the phase plane method as the basis for stability judgment, the longitudinal vehicle speed vx, the road adhesion coefficient μ and the front wheel angle δf are the three most important factors affecting the stability area, and it is necessary to determine the range of variation of the three variables vx, μ and δf as well as the sampling points. In order to reduce the sampling points of the variables, the change step size of the three variables is determined by analyzing the variation law of the sideslip angle of center of mass balance point with the three variables, where the range of the road adhesion coefficient is 0~1.0 , the change step is 0.2, and the pavement adhesion coefficient is divided into 5 sample points: 0.2, 0.4, 0.6, 0.8, 1.0. Therefore, this paper also adopts the same method to divide the pavement adhesion coefficient, and divides the adhesion coefficient into five scope.
Point 6: What are the labels of axes in figure 16?
Response 6: We appreciate your comment. The labels of the axes in Figure 16 are: x-axis: time t/s, y-axis: motor torque/N m, we have corrected the axis labels in Figure 16.
Point 7: In line 485, the difference in the peak value of the sideslip angle compared with an uncontrolled vehicle is 24%, authors should clarify how much this number is for the yaw rate. How much is the difference between the two controllers? Pure motor controller and electrohydraulic coordinated controller?
Response 7: We appreciate your comment. We have carefully checked and corrected some errors in the whole paper. The ambiguities in this paragraph have been corrected accordingly. Specifically: However, the vehicle with the pure motor control and the electrohydraulic coordination control are closer to the ideal value of 0.28 rad/s and 0.032 rad. Among them, the peak values of the pure motor control are about 0.32rad/s and 0.035rad. Compared with the pure motor control, the electro-hydraulic coordinated control reduced by about 0.018rad/s and 0.002rad. And the same time, compared with no control, the peak yaw rate of the ve-hicle with control is reduced by 0.10rad/s, an increase of about 24%, and the peak of the sideslip angle of the center of mass is reduced by 0.012 rad and improved by 27%. Obviously, the yaw rate and the sideslip angle of the center of mass curve of the controlled ve-hicle have a small change range, which is consistent with the ideal value curve, and can track the ideal value curve well. But the yaw rate curve and the sideslip angle of the center of mass curve of the electro-hydraulic coordinated control are closer to the ideal curve, and the control effect is good. We have corrected the corresponding parts in section 5 of the manuscript
Point 8: In line 529, "... 27% higher than that of not applying control ...", it is higher or lower?
Response 8: We appreciate your comment. We have carefully checked and corrected some errors in the whole paper. For example "... 27% higher than that of not applying control ..., it is higher or lower. ". Therefore, for this issue, we have modified this paragraph accordingly. Through the vehicle model establishment, motor parameterization matching, yaw moment control, torque distribution control and joint simulation. The yaw rate and the sideslip angle of the center of mass of the vehicle controlled by electro-hydraulic coordination are smaller than the output parameters of the vehi-cle without control applied. And the yaw rate of the vehicle can better track the ideal yaw rate, and the sideslip angle of center of mass can be kept in a small range and improved by about 27%, improving the vehicle's ability to follow the desired path. We have corrected the corresponding parts in section 7 of the manuscript.
Point 9: Why did the authors select the road adhesion coefficient equal to 0.7 for stability simulation analysis? It is suggested to have the control system tested for some other road adhesion coefficients too as this parameter plays a decisive role in the yaw stability criteria.
Response 9: We appreciate your comment. I think your suggestion is very effective. The reason for choosing the road adhesion coefficient of 0.7 is that most driving environment is dry, and good asphalt or concrete roads, so I did an experiment under this adhesion coefficient. No experiment has been carried out for the adhesion coefficient of wet cement pavement of 0.6. Therfore, this revised paper tested the double-shift condition with the adhesion coefficient of 0.6. We have added the corresponding parts in section 5 of the manuscript
Point 10: Some typos: line 78: propose ====> proposed; line 150: is ===> was; line 320: control controller ====> controller
Response 10: We appreciate your comment. We have carefully checked and corrected some typos in the whole paper. We have corrected " propose " to " proposed ", have corrected " is " to " was " and have corrected " control controller " to " controller ".
Reviewer 3 Report
Article Research on direct yaw moment control of electric vehicles 2 based on electrohydraulic joint actions very actual. I think, that topic is very interesting.
I purpose improvements in followings:
1. Abstract
Please add main aim of this article and describe research methods, which were used.
Describe limitations of your research.
2. Introduction
Is very good written, but I recommend describe similar practical researches, which were provided by others authors. I think, that it may be very beneficial for your article.
3. research
Table 1, please use the some number of decimal. (e.g. 10.1; 20.2 etc.)
I did not found, what is main aim of your article ? Which research methods, were used? Which data was used? Please describe methodology, research and data which were used. Main aim must be defined in your article.
4. Discussion
please add this chapter. Describe unique of your research, limitations of research, new trends, benefits of your research, and establish next researches, which may be provided.
5. Conclusion
Was main aim of this article fulfilled?
I think that article is very good written, but some formal requirements are needed.
Author Response
Response to Reviewer Comments
Dear Reviewer,
The manuscript “Research on direct yaw moment control of electric vehicles based on electrohydraulic joint action” has been revised. Firstly, we would like to express our gratitude to you for us to improve the quality of this manuscript. According to the comments, we have revised the paper, in which the modifications are highlighted in red color.
Best regards,
Taofeng Yan
Point 1: Abstract: Please add main aim of this article and describe research methods, which were used. Describe limitations of your research.
Response 1: We appreciate your comment. We have carefully checked and supplemented some flaws in the abstract of the paper. The main purpose of this paper : in order to solve the problem of lateral instability of the vehicle caused by insufficient lateral force of the tire due to the motor cannot provide enough torque for the tire when the vehicle turns sharply or avoids obstacles in an emergency.
The methods of this paper: a two-degree-of-freedom vehicle dynamics model is established as the reference model, and the B-Class Hatchback vehicle model in the Carsim software is selected as the complete vehicle model of the control system. Secondly, this paper combines the phase plane method of the sideslip angle of the center of mass - the sideslip angle velocity of the center of mass and the threshold method to evaluate the stable state of the vehicle, then uses the sliding mode control algorithm to calculate the additional yaw moment required by the vehicle, and finally realizes the distribution of the torque through the electro-hydraulic coordinated control method—Based on the torque distribution rule of real-time load transfer, and calculate the torque corresponding to the control wheel, the torque generated by the hub motor is transmitted to the wheel. When the output torque of the motor is insufficient, the hydraulic brake is used as a compensation control to redistribute the required torque. However, due to the limitation of experimental equipment, the proposed method could not be applied to the real vehicle test. The real vehicle test can better test the control effect of the proposed method.
Point 2: Introduction: Is very good written, but I recommend describe similar practical researches, which were provided by others authors. I think, that it may be very beneficial for your article.
Response 2: We appreciate your comment. We have revised the Introduction section,layered the original literature review into a summary, and aligned with our own research goals. By describing the similar actual research provided by other authors, analyzing the shortcomings of their research, and proposing our own research content and methods.
Point 3: Researsh: Table 1, please use the some number of decimal. (e.g. 10.1; 20.2 etc.). I did not found, what is main aim of your article ? Which research methods, were used? Which data was used? Please describe methodology, research and data which were used. Main aim must be defined in your article.
Response 3: We appreciate your comment. We have carefully checked and corrected tabular data in the whole paper. Such as Table 1. The main purpose of this paper : in order to solve the problem of lateral instability of the vehicle caused by insufficient lateral force of the tire due to the motor cannot provide enough torque for the tire when the vehicle turns sharply or avoids obstacles in an emergency.
The methods of this paper: a two-degree-of-freedom vehicle dynamics model is established as the reference model, and the B-Class Hatchback vehicle model in the Carsim software is selected as the complete vehicle model of the control system. Secondly, this paper combines the phase plane method of the sideslip angle of the center of mass - the sideslip angle velocity of the center of mass and the threshold method to evaluate the stable state of the vehicle, then uses the sliding mode control algorithm to calculate the additional yaw moment required by the vehicle, and finally realizes the distribution of the torque through the electro-hydraulic coordinated control method—Based on the torque distribution rule of real-time load transfer, and calculate the torque corresponding to the control wheel, the torque generated by the hub motor is transmitted to the wheel. When the output torque of the motor is insufficient, the hydraulic brake is used as a compensation control to redistribute the required torque.
Data used in this paper: the main parameters of the vehicle model used in this paper are derived from the Jiangling New Energy E180 autonomous vehicle, as shown in Table 1, and according to the dynamic performance requirements of the vehicle, as shown in Table 2, the basic parameters of the motor are calculated, as shown in Table 3, and the Simulink modeling. The stability domain boundary parameters C1 and C2 used in stability judgment, combined with the road adhesion coefficient, are obtained by the calculation formula of the phase plane method, and the yaw angular velocity limit value C3 refers to the literature [28]. as shown in Table 4.
Point 4: Discussion: please add this chapter. Describe unique of your research, limitations of research, new trends, benefits of your research, and establish next researches, which may be provided.
Response 4: We appreciate your comment. We have added this chapter. It mainly describes the uniqueness and benefits of the research in this paper, and points out the limitations of the research in this paper, and proposes the direction of future research. We have added the corresponding parts in Section 6 of the manuscript.
Point 5: Conclusion: Was main aim of this article fulfilled?
Response 5: We appreciate your comment. The purpose of this paper is achieved. From the results of simulation verification, it can be seen that the electro-hydraulic coordinated control effect is better. When the output torque of the motor is limited, the hydraulic brake compensation control system intervenes in time to perform auxiliary braking, so that the vehicle can turn in time and continue to drive stably. The electro-hydraulic coordinated control can provide sufficient demand torque to solve the problems of insufficient control torque and vehicle instability when the vehicle is turning, and the conclusion part is modified.

Round 2
Reviewer 1 Report
The reviewer suggests that this paper is suitable for publication in Sustainability Journal.
Reviewer 3 Report
I think, that article is very good written.
I recommend accept this article in current version.